# Neural Contractive Dynamical Systems

**Hadi Beik-Mohammadi**[1,2], **Søren Hauberg**[3], **Georgios Arvanitidis**[3], **Nadia Figueroa**[4],
**Gerhard Neumann**[2], and **Leonel Rozo**[1]
[1] Bosch Center for Artificial Intelligence (BCAI), [2] Karlsruhe Institute of Technology (KIT),
[3] Technical University of Denmark (DTU), [4] University of Pennsylvania (UPenn).
Emails: [hadi.beik-mohammadi, leonel.rozo]@de.bosch.com, [sohau, gear]@dtu.dk,
nadiafig@seas.upenn.edu, gerhard.neumann@kit.edu

## Abstract

Stability guarantees are crucial when ensuring that a fully autonomous robot does not take undesirable or potentially harmful actions. Unfortunately, global stability guarantees are hard to provide in dynamical systems learned from data, especially when the learned dynamics are governed by neural networks. We propose a novel methodology to learn *neural contractive dynamical systems*, where our neural architecture ensures contraction, and hence, global stability. To efficiently scale the method to high-dimensional dynamical systems, we develop a variant of the variational autoencoder that learns dynamics in a low-dimensional latent representation space while retaining contractive stability after decoding. We further extend our approach to learning contractive systems on the Lie group of rotations to account for full-pose end-effector dynamic motions. The result is the first highly flexible learning architecture that provides contractive stability guarantees with capability to perform obstacle avoidance. Empirically, we demonstrate that our approach encodes the desired dynamics more accurately than the current state-of-the-art, which provides less strong stability guarantees.

## 1 Stability in Robot Learning

Deploying a fully autonomous robot requires guarantees of stability to ensure that the robot does not perform unwanted, potentially dangerous, actions. Consider the robot in Fig. 1, which is trained from demonstrations (black dots) to execute a sinusoidal motion (orange). If the robot is pushed away (red perturbation) from its planned trajectory, then it should still be guaranteed to reproduce a motion similar to the demonstration pattern (blue). Manually designed robot movements could achieve such stability guarantees, but even skilled engineers struggle to hand-code highly dynamic motions (Billard et al., 2016). In contrast, learning robot dynamics from demonstrations has shown to be an efficient and intuitive approach for encoding highly dynamic motions into a robot's repertoire (Schaal et al., 2003). Unfortunately, these learning-based approaches often struggle to ensure stability as they rely on the machine learning model to extrapolate in a con-

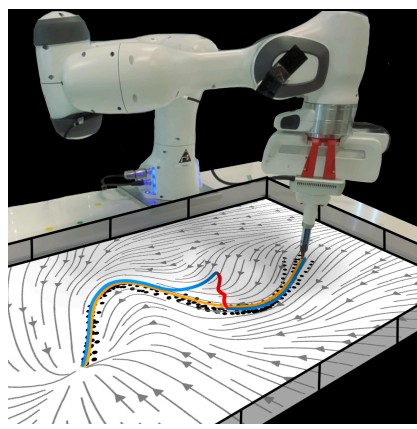

Figure 1: Robot motion executed via a neural contractive dynamical system (NCDS).

trolled manner. In particular, it has turned out to be surprisingly difficult to control the extrapolating behavior of neural network models (Xu et al., 2021), which hampers stability guarantees.

**Stability is commonly ensured** through *asymptotic* or *contraction* guarantees. Asymptotic stability ensures that all motions converge to a fixed *point* (known as the system's *attractor*) (Khansari-Zadeh & Billard, 2011; 2014; Neumann & Steil, 2015; Zhang et al., 2022b;a). This is suitable when the only requirement is that the robot eventually reaches a certain configuration, e.g. its end-effector is at a specific goal position. Many tasks, however, require the robot to dynamically follow desired trajectories, e.g. in flexible manufacturing, human-robot interaction, or in entertainment settings.

In these cases, asymptotic stability is insufficient. A stronger notion of stability is provided by *contraction theory* (Lohmiller & Slotine, 1998), which ensures that all the path integrals regardless of their initial state incrementally converge over time. Unfortunately, the mathematical requirements of a contractive system are difficult to ensure in popular neural network architectures.

**Existing contractive learning methods** focus on low-capacity models fitted under contraction constraints. Ravichandar et al. (2017) and Ravichandar & Dani (2019) both use Gaussian mixture models to encode the demonstrations and model the dynamical system with Gaussian mixture regression (GMR), which is fitted under contraction constraints. Blocher et al. (2017) also use GMR to model the system dynamics but instead include a stabilizing control input that guarantees a local contractive behavior. Likewise, Sindhwani et al. (2018) model the dynamical system with a well-crafted kernel, under which optimization remains convex w.r.t. contraction constraints. Note that the above works impose the contraction constraints during optimization, which complicates model training.

Developing high-capacity (deep) models is difficult as off-the-shelf neural network architectures provide no stability guarantees. Instead, current approaches split the task into two steps: first, learn a non-contractive dynamical system, and then estimate a Riemannian metric (Sec. 2.1), parametrized by a neural network, under which the system becomes contractive (Tsukamoto et al., 2021a; Dawson et al., 2023). The approach, thus, requires training two separate networks, which is impractical and comes with additional challenges. For example, Sun et al. (2020) learns the metric by regularizing towards a contractive system, such that stability can *only* be ensured near training data. Alternatively, Tsukamoto & Chung (2021) develop a convex optimization problem for learning the metric from sampled *optimal* metrics. Unfortunately, accessing such optimal metrics is a difficult problem in itself. Additionally, Kozachkov et al. (2022) introduce a network of small and simple recurrent neural networks with built-in contractive behavior as an alternative to a single, larger network.

**In this paper**, we propose the *neural contractive dynamical system (NCDS)* which is the first neural network architecture that is guaranteed to be contractive for all parameter values. This implies that NCDS can easily be incorporated into existing pipelines to provide stability guarantees and then trained using standard unconstrained optimization. As demonstration data may be high dimensional, we secondly propose an injective variant of the variational autoencoder that allows learning low-dimensional latent dynamics, while ensuring contraction guarantees for the *decoded* dynamical system. Thirdly, as most complex robotic tasks involve full-pose end-effector movements (including *orientation* dynamics), we extend our approach to cover the Lie group $\mathcal{SO}(3)$. Finally, we show that NCDS can trivially be modified to avoid (dynamic) obstacles without sacrificing stability guarantees.

## 2 LEARNING CONTRACTIVE VECTOR FIELDS

Our ambition is a flexible neural architecture, which is guaranteed to always be contractive. We, however, first introduce contraction theory as this is a prerequisite for our design.

### 2.1 BACKGROUND: CONTRACTIVE DYNAMICAL SYSTEMS

To begin with, assume an autonomous dynamical system $\dot{\boldsymbol{x}}_t = f(\boldsymbol{x}_t)$, where $\boldsymbol{x}_t \in \mathbb{R}^D$ is the state variable, $f : \mathbb{R}^D \to \mathbb{R}^D$ is a $C^1$ function and $\dot{\boldsymbol{x}}_t = {}^{\mathrm{d}\boldsymbol{x}}/\mathrm{d}t$ denotes temporal differentiation. As illustrated in Fig. 1, contraction stability ensures that all solution trajectories of a nonlinear system $f$ incrementally converge regardless of initial conditions $\boldsymbol{x}_0, \dot{\boldsymbol{x}}_0$, and temporary perturbations (Lohmiller & Slotine, 1998). The system stability can, thus, be analyzed differentially, i.e. we can ask if two nearby trajectories converge to one another. Specifically, contraction theory defines a measure of distance between neighboring trajectories, known as the *contraction metric*, in which the distance decreases exponentially over time (Jouffroy & I. Fossen, 2010; Tsukamoto et al., 2021b).

Formally, an autonomous dynamical system yields the differential relation $\delta\dot{\boldsymbol{x}} = \boldsymbol{J}(\boldsymbol{x})\delta\boldsymbol{x}$, where $\boldsymbol{J}(\boldsymbol{x}) = {}^{\partial f}/\partial\boldsymbol{x}$ is the system Jacobian and $\delta\boldsymbol{x}$ is a virtual displacement (i.e., an infinitesimal spatial displacement between the nearby trajectories at a fixed time). Note that we have dropped the time index $t$ to limit notational clutter. The rate of change of the corresponding infinitesimal squared distance $\delta\boldsymbol{x}^\top\delta\boldsymbol{x}$ is then,

$$\frac{\mathrm{d}}{\mathrm{d}t}(\delta\boldsymbol{x}^\top\delta\boldsymbol{x}) = 2\delta\boldsymbol{x}^\top\delta\dot{\boldsymbol{x}} = 2\delta\boldsymbol{x}^\top\boldsymbol{J}(\boldsymbol{x})\delta\boldsymbol{x}. \tag{1}$$

It follows that if the symmetric part of the Jacobian $\boldsymbol{J}(\boldsymbol{x})$ is negative definite, then the infinitesimal squared distance between neighboring trajectories decreases over time. This is formalized as:

**Definition 1 (Contraction stability (Lohmiller & Slotine, 1998))** *An autonomous dynamical system $\dot{x} = f(x)$ exhibits a contractive behavior if its Jacobian $J(x) = \partial f / \partial x$ is negative definite, or equivalently if its symmetric part is negative definite. This means that there exists a constant $\tau > 0$ such that $\delta x^\top \delta x$ converges to zero exponentially at rate $2\tau$. This can be summarized as,*

$$\exists\, \tau > 0 \quad s.t. \quad \forall x, \quad \frac{1}{2}\Big(J(x) + J(x)^\top\Big) \prec -\tau I \prec 0. \tag{2}$$

The above analysis can be generalized to account for a more general notion of distance of the form $\delta x^\top M(x)\delta x$, where $M(x)$ is a positive-definite matrix known as the (Riemannian) contraction metric (Tsukamoto et al., 2021b; Dawson et al., 2023). In our work, we learn a dynamical system $f$ so that it is inherently contractive since its Jacobian $J(x)$ fulfills the condition given in Eq. 2. Consequently, it is not necessary to specifically learn a contraction metric as it corresponds to an identity matrix. For our purposes, we, thus, define contraction guarantees following Eq. 2.

## 2.2 NEURAL CONTRACTIVE DYNAMICAL SYSTEMS (NCDS)

From Definition 1, we seek a flexible neural network architecture, such that the symmetric part of its Jacobian is negative definite. We consider the dynamical system $\dot{x} = f(x)$, where $x \in \mathbb{R}^D$ denotes the system's state and $f : \mathbb{R}^D \to \mathbb{R}^D$ is parametrized by a neural network. It is not obvious how to impose a negative definiteness constraint on a network's Jacobian without compromising its expressiveness. To achieve this, we first design a neural network $\hat{J}_f$ representing the Jacobian of our final network. This produces matrix-valued negative definite outputs. The final neural network will then be formed by integrating the Jacobian network. Specifically, we define the Jacobian as,

$$\hat{J}_f(x) = -(J_\theta(x)^\top J_\theta(x) + \epsilon \mathbb{I}_D), \tag{3}$$

where $J_\theta : \mathbb{R}^D \to \mathbb{R}^{D \times D}$ is a neural network parameterized by $\theta$, $\epsilon \in \mathbb{R}^+$ is a small positive constant, and $\mathbb{I}_D$ is an identity matrix of size $D$. Intuitively, $J_\theta$ can be interpreted as the (approximate) square root of $\hat{J}_f$. Clearly, $\hat{J}_f$ is negative definite as all eigenvalues are bounded from above by $-\epsilon$.

Next, we take inspiration from Lorraine & Hossain (2019) and integrate $\hat{J}_f$ to produce a function $f$, which is implicitly parametrized by $\theta$, and has Jacobian $\hat{J}_f$. The fundamental theorem of calculus for line integrals tells us that we can construct such a function by a line integral of the form

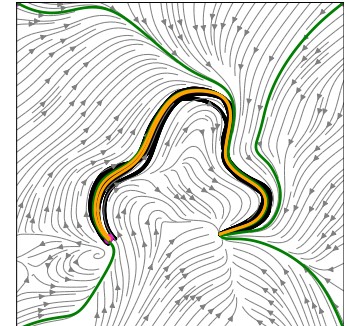

$$\dot{x} = f(x) = \dot{x}_0 + \int_0^1 \hat{J}_f\left(c\left(x, t, x_0\right)\right) \dot{c}(x, t, x_0)\mathrm{d}t, \tag{4}$$

$$c(x, t, x_0) = (1 - t)\, x_0 + t x \quad \text{and} \quad \dot{c}(t, x_0, x) = x - x_0, \tag{5}$$

where $x_0$ and $\dot{x}_0 = f(x_0)$ represent the initial conditions of the state variable and its first-order time derivative, respectively. The input point $x_0$ can be chosen arbitrarily (e.g. as the mean of the training data or it can be learned), while the corresponding function value $\dot{x}_0$ has to be estimated alongside the parameters $\theta$. Therefore, given a set of demonstrations denoted as $\mathcal{D} = \{x_i, \dot{x}_i\}$, our objective is to learn a set of parameters $\theta$ along with the initial conditions $x_0$ and $\dot{x}_0$, such that the integration in Eq. 4 enables accurate reconstruction of the velocities $\dot{x}_i$ given the state $x_i$.

Figure 2: The learned vector field (grey) and demonstrations (black). Yellow and green trajectories show path integrals starting from demonstration starting points and random points, respectively.

The integral in Eq. 4 is similar to neural ordinary differential equations (Chen et al., 2018b), with the subtle difference that it is a second-order equation as the outcome pertains to the *velocity* at state $x$. We can, thus, view this system as a second-order neural ordinary equation (Norcliffe et al., 2021) and solve it using off-the-shelf numerical integrators. The resulting function $f$ will have a negative definite Jacobian for any choice of $\theta$ and is consequently contractive by construction. In other words, we can control the extrapolation behavior of the neural network that parametrizes our dynamical system $f$ via the negative-definiteness of $\hat{J}_f(x)$. We call this the *neural contractive dynamical system (NCDS)*, and emphasize that this approach has two key benefits: *(1)* we can use any smooth neural network $J_\theta$ as our base model, and *(2)* training can then be realized using ordinary *unconstrained* optimization, unlike previous approaches. Figure 2 shows an example NCDS vector field, which is clearly highly flexible even if it guarantees contractive stability.

## 3 LEARNING LATENT CONTRACTIVE DYNAMICS

Learning highly nonlinear contractive dynamical systems in high-dimensional spaces is difficult. These systems may exhibit complex trajectories with intricate interdependencies among the system variables, making it challenging to capture the underlying dynamics. In the experiments, we show that NCDS works very well for low-dimensional problems, but when only limited data is available, the approach becomes brittle in higher dimensions. A common approach in such cases is to first reduce the data dimensionality, and work as before in the resulting low-dimensional latent space (Chen et al., 2018a; Hung et al., 2022; Beik-Mohammadi et al., 2021). The main difficulty is that even if the latent dynamics are contractive, the associated high-dimensional dynamics need not be. This we solve next.

### 3.1 BACKGROUND: DEEP GENERATIVE MODELS

**Variational autoencoders (VAEs).** The main goal of *deep generative models* is to approximate the true underlying probability density $p(\boldsymbol{x})$ given a finite set of training data in an ambient space $\mathcal{X}$, by considering a lower-dimensional latent space $\mathcal{Z}$. In particular, the *variational autoencoder (VAE)* (Kingma & Welling, 2014; Rezende et al., 2014) is a latent variable model, often specified through a prior $p(\boldsymbol{z}) = \mathcal{N}(\boldsymbol{z} \mid \boldsymbol{0}, \mathbb{I}_d)$ where $\boldsymbol{z} \in \mathcal{Z}$, and a likelihood $p_{\boldsymbol{\phi}}(\boldsymbol{x}|\boldsymbol{z}) = \mathcal{N}(\boldsymbol{x} \mid \mu_{\boldsymbol{\phi}}(\boldsymbol{z}), \mathbb{I}_D \sigma_{\boldsymbol{\phi}}^2(\boldsymbol{z}))$ with $\boldsymbol{x} \in \mathcal{X}$. Typically, the mean and the variance of the likelihood are parametrized using deep neural networks $\mu_{\boldsymbol{\phi}} : \mathcal{Z} \to \mathcal{X}$ and $\sigma_{\boldsymbol{\phi}}^2 : \mathcal{Z} \to \mathbb{R}_+^D$ with parameters $\boldsymbol{\phi}$, and $\mathbb{I}_D$ and $\mathbb{I}_d$ being identity matrices of size $D$ and $d$, respectively. These neural networks are trained by maximizing the evidence lower bound (ELBO) (Kingma & Welling, 2014). The latent variable $\boldsymbol{z}$ is approximated using a variational encoder $\mu_{\boldsymbol{\xi}}(\boldsymbol{x})$ and is often interpreted as the low-dimensional representation of an observation $\boldsymbol{x}$. In our work, we use a VAE to provide low-dimensional representations of individual points along observed trajectories in order to learn a latent contractive dynamical system.

**Injective flows.** A limitation of VAEs is that their marginal likelihood is intractable and we have to rely on a bound. When $\dim(\mathcal{X}) = \dim(\mathcal{Z})$, we can apply the change-of-variables theorem to evaluate the marginal likelihood exactly, giving rise to *normalizing flows* (Tabak & Turner, 2013). This requires the decoder to be diffeomorphic, i.e. a smooth invertible function with a smooth inverse. In order to extend this to the case where $\dim(\mathcal{X}) > \dim(\mathcal{Z})$, Brehmer & Cranmer (2020) proposed an *injective flow*, which implements a zero-padding operation (see Sec.3.2 and Equation 6) on the latent variables alongside a diffeomorphic decoder, such that the resulting function is injective.

### 3.2 LATENT NEURAL CONTRACTIVE DYNAMICAL SYSTEMS

As briefly discussed above, we want to reduce data dimensionality with a VAE, but we further require that any latent contractive dynamical system remains contractive after it has been decoded into the data space. To do so, we can leverage the fact that contraction is invariant under coordinate changes (Manchester & Slotine, 2017; Kozachkov et al., 2023). This means that the transformation between the latent and data spaces may be generally achieved through a diffeomorphic mapping.

**Theorem 1 (Contraction invariance under diffeomorphisms (Manchester & Slotine, 2017))**
*Given a contractive dynamical system $\dot{\boldsymbol{x}} = f(\boldsymbol{x})$ and a diffeomorphism $\psi$ applied on the state $\boldsymbol{x} \in \mathbb{R}^D$, the transformed system preserves contraction under the change of coordinates $\boldsymbol{y} = \psi(\boldsymbol{x})$. Equivalently, contraction is also guaranteed under a differential coordinate change $\delta_{\boldsymbol{y}} = \frac{\partial \psi}{\partial \boldsymbol{x}} \delta_{\boldsymbol{x}}$.*

Following Theorem 1, we learn a VAE with an injective decoder $\mu : \mathcal{Z} \to \mathcal{X}$. Letting $\mathcal{M} = \mu(\mathcal{Z})$ denote the image of $\mu$, then $\mu$ is a diffeomorphism between $\mathcal{Z}$ and $\mathcal{M}$, such that Theorem 1 applies. Geometrically, $\mu$ spans a $d$-dimensional submanifold of $\mathcal{X}$ on which the dynamical system operates.

Here we leverage the zero-padding architecture from Brehmer & Cranmer (2020) for the decoder. Formally, an injective flow $\mu : \mathcal{Z} \to \mathcal{X}$ learns an injective mapping between a low-dimensional latent space $\mathcal{Z}$ and a higher-dimensional data space $\mathcal{X}$. Injectivity of the flow ensures that there are no singular points or self-intersections in the flow, which may compromise the stability of the system dynamics in the data space. The injective decoder $\mu$ is composed of a zero-padding operation on the latent variables followed by a series of $K$ invertible transformations $g_k$. This means that,

$$\mu = g_K \circ \cdots \circ g_1 \circ \mathrm{Pad}, \tag{6}$$

where $\mathrm{Pad}(\boldsymbol{z}) = [z_1 \cdots z_d \; 0 \cdots 0]^\top$ represents a $D$-dimensional vector $\boldsymbol{z}$ with additional $D-d$ zeros. We emphasize that this decoder is an injective mapping between $\mathcal{Z}$ and $\mu(\mathcal{Z}) \subset \mathcal{X}$, such that a decoded contractive dynamical system remains contractive.

Specifically, we propose to learn a latent data representation using a VAE, where the decoder mean $\mu_{\boldsymbol{\xi}}$ follows the architecture in Eq. 6. Empirically, we have found that training stabilizes when the variational encoder takes the form $q_{\boldsymbol{\xi}}(\boldsymbol{z}|\boldsymbol{x}) = \mathcal{N}(\boldsymbol{z} \mid \mu_{\widetilde{\boldsymbol{\xi}}}^{-1}(\boldsymbol{x}), \mathbb{I}_d \sigma_{\widetilde{\boldsymbol{\xi}}}^2(\boldsymbol{x}))$, where $\mu_{\widetilde{\boldsymbol{\xi}}}^{-1}$ is the approximate inverse of $\mu_{\boldsymbol{\xi}}$ given by,

$$\mu_{\widetilde{\boldsymbol{\xi}}}^{-1} = \mathrm{Unpad} \circ g_1^{-1} \circ \cdots \circ g_K^{-1}, \tag{7}$$

where $\mathrm{Unpad} : \mathbb{R}^D \to \mathbb{R}^d$ removes the last $D-d$ dimensions of its input as an approximation to the inverse of the zero-padding operation. We emphasize that an exact inverse is not required to evaluate a lower bound of the model evidence.

It is important to note that the state $\boldsymbol{x}$ solely encodes the positional information of the system, disregarding the velocity $\dot{\boldsymbol{x}}$. In order to decode the latent velocity $\dot{\boldsymbol{z}}$ into the data space velocity $\dot{\boldsymbol{x}}$, we exploit the Jacobian matrix associated with the decoder mean function $\mu_{\boldsymbol{\xi}}$, computed as $\boldsymbol{J}_{\mu_{\boldsymbol{\xi}}}(\boldsymbol{z}) = \partial \mu_{\boldsymbol{\xi}}/\partial \boldsymbol{z}$. This enables the decoding process formulated as below,

$$\dot{\boldsymbol{x}} = \boldsymbol{J}_{\mu_{\boldsymbol{\xi}}}(\boldsymbol{z})\dot{\boldsymbol{z}}. \tag{8}$$

The above tools let us learn a contractive dynamical system on the latent space $\mathcal{Z}$, where the contraction is guaranteed by employing the NCDS architecture (Sec. 2.2). Then, the latent velocities[1] $\dot{\boldsymbol{z}}$ given by such a contractive dynamical system can be mapped to the data space $\mathcal{X}$ using Eq. 8. Assuming the initial robot configuration $\boldsymbol{x}_0$ is in $\mathcal{M}$ (i.e., $\boldsymbol{x}_0 = f(\boldsymbol{z}_0)$), the subsequent motion follows a contractive system along the manifold. If the initial configuration $\boldsymbol{x}_0$ is not in $\mathcal{M}$, the encoder is used to approximate a projection onto $\mathcal{M}$, producing $\boldsymbol{z}_0 = \mu^{\sim 1}(\boldsymbol{x}_0)$. Note that the movement from $\boldsymbol{x}_0$ to $f(\boldsymbol{z}_0)$ need not be contractive, but this is a finite-time motion, after which the system is contractive.

### 3.3 Learning position and orientation dynamics

So far, we have focused on Euclidean robot states, but in practice, the end-effector motion also involves rotations, which do not have an Euclidean structure. We first review the group structure of rotation matrices and then extend NCDS to handle non-Euclidean data using Theorem 1.

**Orientation parameterization.** Three-dimensional spatial orientations can be represented in several ways, including Euler angles, unit quaternions, and rotation matrices (Shuster, 1993). We focus on the latter. The set of rotation matrices forms a Lie group, known as the *special orthogonal group* $\mathcal{SO}(3) = \left\{ \boldsymbol{R} \in \mathbb{R}^{3 \times 3} \mid \boldsymbol{R}^\top \boldsymbol{R} = \mathbb{I}, \det(\boldsymbol{R}) = 1 \right\}$. Every Lie group is associated with its Lie algebra, which represents the tangent space at its origin (Fig. 3). This Euclidean tangent space allows us to operate with elements of the group via their projections on the Lie algebra (Solà et al., 2018). In the context of $\mathcal{SO}(3)$, its Lie algebra $\mathfrak{so}(3)$ is the set of all $3 \times 3$ skew-symmetric matrices $[\boldsymbol{r}]_\times$. This skew-symmetric matrix exhibits three degrees of freedom, which can be reparameterized as a 3-dimensional vector $\boldsymbol{r} = [r_x, r_y, r_z] \in \mathbb{R}^3$.

We can map back and forth between the Lie group $\mathcal{SO}(3)$ and its associated Lie algebra $\mathfrak{so}(3)$ using the *logarithmic* and *exponential maps*, denoted $\mathrm{Log} : \mathcal{SO}(3) \to \mathfrak{so}(3)$ and $\mathrm{Exp} : \mathfrak{so}(3) \to \mathcal{SO}(3)$, respectively. We provide explicit formulae for these in Appendix A.2. Due to wrapping (e.g. $360°$ rotation corresponds to $0°$), the exponential map is surjective. This implies that the inverse, i.e. $\mathrm{Log}$, is multivalued, which complicates matters. However, for vectors $\boldsymbol{r} \in \mathfrak{so}(3)$, both $\mathrm{Exp}$ and $\mathrm{Log}$ are diffeomorphic if $\|\boldsymbol{r}\| < \pi$ (Falorsi et al., 2019; Urain et al., 2022). This $\pi$-ball $\mathcal{B}_\pi$ is known as the *first cover* of the Lie algebra and corresponds to the part where no wrapping occurs.

Figure 3: Aspects of the Lie group $\mathcal{SO}(3)$.

**NCDS on Lie groups.** Consider the situation where the system state is a rotation, i.e. $\boldsymbol{x} \in \mathcal{SO}(3)$, which we seek to model with a latent NCDS. From a generative point of view, we first construct a decoder $\mu : \mathcal{Z} \to \mathfrak{so}(3)$ with outputs in the Lie algebra. We can then apply the exponential map to generate a rotation matrix $\boldsymbol{R} \in \mathcal{SO}(3)$, such that the complete decoder becomes $\mathrm{Exp} \circ \mu$.

---

[1] For training, the latent velocities are simply estimated by a numerical differentiation w.r.t the latent state $\boldsymbol{z}$.

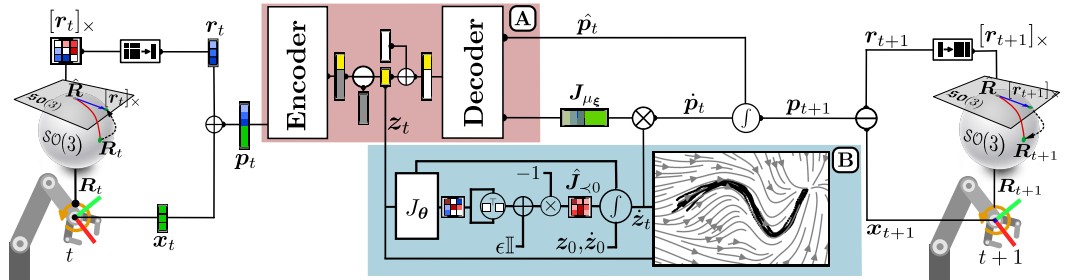

Figure 4: Architecture overview: a single iteration of NCDS simultaneously generating position and orientation dynamics. **(A) VAE (pink box):** The encoder processes the concatenated position-orientation data $\boldsymbol{p}_t$, yielding a resulting vector that is subsequently divided into two components: the latent code $\boldsymbol{z}$ (yellow squares) and the surplus (gray squares). The $\mathrm{Unpad}$ function in Equation 7 removes the unused segment. The unpadded latent code $\boldsymbol{z}$ is fed to the contraction module and simultaneously padded with zeros (white squares) before being passed to the injective decoder. **(B) Contraction (blue box):** The Jacobian network output, given the latent codes, is reshaped into a square matrix and transformed into a negative definite matrix using Equation 2. The numerical integral solver then computes the latent velocity $\dot{\boldsymbol{z}}$. Later, using Eq. 8, $\dot{\boldsymbol{z}}$ is mapped to the input-space velocity via the decoder's Jacobian $\boldsymbol{J}_{\mu_{\boldsymbol{\xi}}}$.

Unfortunately, even if $\mu$ is injective, we cannot ensure that $\mathrm{Exp} \circ \mu$ is also injective (since $\mathrm{Exp}$ is surjective), which then breaks the stability guarantees of NCDS. Here we leverage the result that $\mathrm{Exp}$ is a diffeomorphism as long as we restrict ourselves to the first cover of $\mathfrak{so}(3)$. Specifically, if we choose a decoder architecture such that $\mu : \mathcal{Z} \to \mathcal{B}_\pi$ is injective and have outputs on the first cover, then $\mathrm{Exp} \circ \mu$ is injective, and stability is ensured (see Appendix A.5). Note that during training, observed rotation matrices can be mapped directly into the first cover of $\mathfrak{so}(3)$ using the logarithmic map. When the system state consists of both rotations and positions, we simply decode to higher dimensional variables and apply exponential and logarithmic maps on the appropriate dimensions. Figure 4 depicts this general form of NCDS.

### 3.4 OBSTACLE AVOIDANCE VIA MATRIX MODULATION

In real-world scenarios, obstacle avoidance is critical to achieving safe autonomous robots. Thus, the learned dynamical system should effectively handle previously unseen obstacles, without interfering with the overall contracting behavior of the system. Fortunately, NCDS can easily be adapted to perform contraction-preserving obstacle avoidance using the dynamic modulation matrix $\boldsymbol{G}$ from Huber et al. (2022). This approach locally reshapes the learned vector field in the proximity of obstacles, while preserving contraction guarantees. To avoid obstacles effectively, it is imperative to know both the position and geometry of the obstacle. This makes obstacle avoidance on the VAE latent space difficult as we need to map both obstacle location and geometry to the latent space $\mathcal{Z}$.

Instead, we directly apply the modulation matrix to the data space $\mathcal{X}$ of the decoded dynamical system. Specifically, Huber et al. (2022) shows how to construct a modulation matrix $\boldsymbol{G}$ from the object location and geometry, such that the vector field $\dot{\boldsymbol{x}} = \boldsymbol{G}(\boldsymbol{x})\boldsymbol{J}_{\mu_{\boldsymbol{\xi}}}(\boldsymbol{z})\dot{\boldsymbol{z}}$ is both contractive and steers around the obstacle. With this minor modification, NCDS can be adapted to avoid obstacles. We provide the details in Appendix A.3. Note that this is tailored for obstacle avoidance at the end-effector level. In order to extend this to multiple-limb obstacle avoidance, the modulation matrix must be refined to incorporate the robot body, and NCDS must be trained using joint space trajectories.

## 4 EXPERIMENTAL RESULTS

To evaluate the efficiency of NCDS, we consider several synthetic and real tasks. Comparatively, we show that NCDS is the only method to scale gracefully to higher dimensional problems, due to the latent structure. We further demonstrate the ability to build dynamic systems on the Lie group of rotations and to avoid obstacles while ensuring stability. Neither of the baseline methods have such capabilities. Experimental details are in Appendix A.4, while ablation studies of NCDS modeling choices are in Appendix A.6.3. Furthermore, the videos are available at: https://sites.google.com/view/neuralcontraction/home.

**Datasets.** There are currently no established benchmarks for contraction-stable robot motion learning, so we focus on two datasets. First, we test our approach on the LASA dataset (Lemme et al., 2015), often used for benchmarking asymptotic stability. This consists of 26 different two-

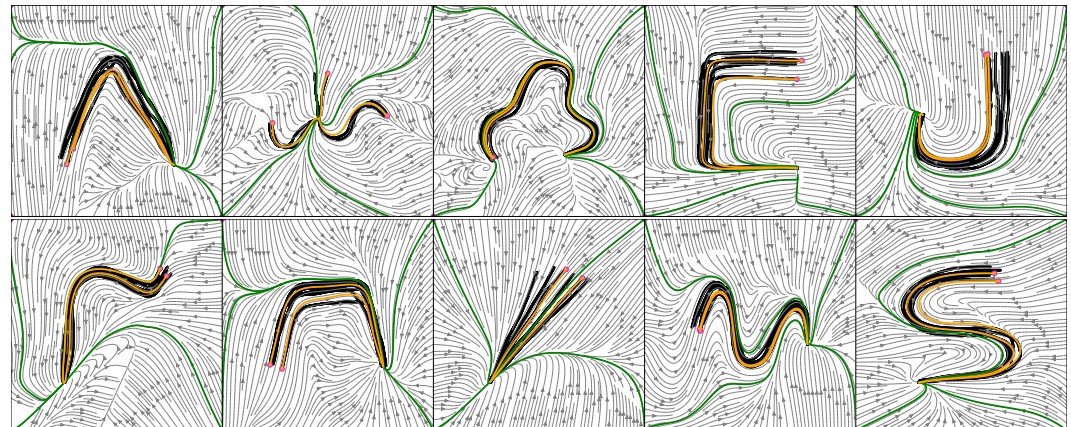

Figure 5: Visualization of the LASA-2D dataset: Gray contours represent the learned vector field, black trajectories depict demonstrations, and orange/green trajectories illustrate path integrals starting from the initial points of the demonstrations and plot corners. The magenta circles indicate the initial points of the path integrals.

dimensional hand-written trajectories, which the robot is tasked to follow. To evaluate our method, we have selected 10 different trajectories from this dataset. To ensure consistency and comparability, we preprocess all trajectories to all stop at the same target state. Additionally, we omit the initial few points of each demonstration, to ensure that the only state exhibiting zero velocity is the target state. To further complicate this easy task, we task the robot to follow 2, or 4 stacked trajectories, resulting in 4 (LASA-4D), or 8-dimensional (LASA-8D) data, respectively. We further consider a dataset consisting of 5 trajectories from a 7-DoF Franka-Emika Panda robot using a Cartesian impedance controller.

**Metrics.** We use dynamic time warping distance (DTWD) as the established quantitative measure of reproduction accuracy w.r.t. a demonstrated trajectory, assuming equal initial conditions (Ravichandar et al., 2017; Sindhwani et al., 2018). This is defined as,

$$\mathrm{DTWD}(\tau_x, \tau_{x'}) = \sum_{j \in l(\tau_{x'})} \min_{i \in l(\tau_x)} \Big( d(\tau_{x_i}, \tau_{x'_j}) \Big) + \sum_{i \in l(\tau_x)} \min_{j \in l(\tau_{x'})} \Big( d(\tau_{x_i}, \tau_{x'_j}) \Big),$$

where $\tau_x$ and $\tau_{x'}$ are two trajectories (e.g. a path integral and a demonstration trajectory), $d$ is a distance function (e.g. Euclidean distance), and $l(\tau)$ is the length of trajectory $\tau$.

**Baseline methods.** Our work is the first contractive neural network architecture, so we cannot compare NCDS directly to methods with identical goals. Instead, we compare to existing methods that provide asymptotic stability guarantees. In particular, *Euclideanizing flow* (Rana et al., 2020), *Imitation flow* (Urain et al., 2020) and *SEDS* (Khansari-Zadeh & Billard, 2011).

### 4.1 COMPARATIVE RESULTS

We first evaluate NCDS on two-dimensional trajectories for ease of visualization. Here data is sufficiently low-dimensional that we do not consider a latent structure. Figure 5 shows the learned vector fields (gray contours) for 10 diverse trajectory shapes chosen according to their difficulty from the LASA dataset, covering a wide range of demonstration patterns (black curves) and dynamics. We observe that NCDS effectively captures and replicates the underlying dynamics. We compute additional path integrals starting from outside the data support (green curves) to assess the generalization capability of NCDS beyond the observed demonstrations. Remarkably, NCDS successfully generates plausible trajectories even in regions not covered by the original demonstration data. Moreover, the yellow path integrals, starting from the initial points of the demonstrations, show that our approach can also reproduce the demonstrated trajectories accurately. This is evidence that the contractive construction is a viable approach to controlling the extrapolation properties of the neural network.

Table 1 shows average DTWD distances among five generated path integrals and five demonstration trajectories for both NCDS and baseline methods. For the two-dimensional problem, both Euclideanizing flow and Imitation flow outperform NCDS and SEDS, though all methods perform quite well. To investigate stability, Fig. 6 displays the average distance over time among five path integrals starting from random nearby initial points for different methods. We observe that only for NCDS does this distance decrease monotonically, which indicates that it is the only contractive method.

NCDS       IF       EF       SEDS

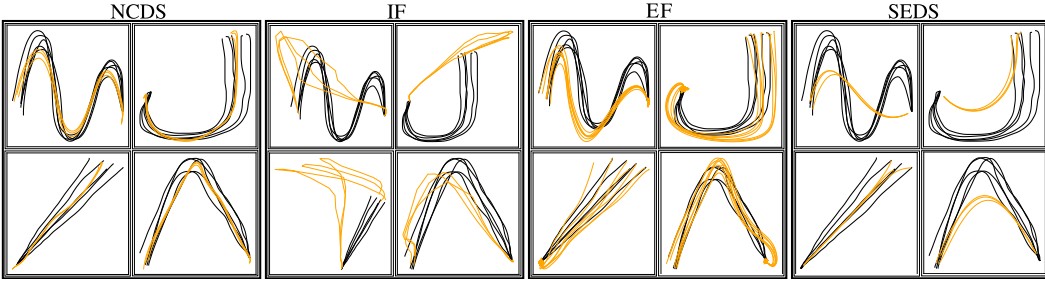

Figure 7: Path integrals generated using different methods on LASA-8D. Black curves are training data, while yellow are the learned path integrals. To construct the 8D dataset, we have concatenated 4 different 2D datasets.

To test how these approaches scale to higher-dimensional settings, we consider the LASA-4D and LASA-8D datasets, where we train NCDS with a two-dimensional latent representation. From Table 1, it is evident that the baseline methods quickly deteriorate as the data dimension increases, and only NCDS gracefully scales to higher dimensional data. This is also evident from Fig. 7 which shows the training data and reconstructions of different LASA-8D dimensions with different methods. These results clearly show the value of having a low-dimensional latent structure (more details in Appendix A.6.5).

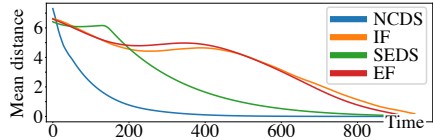

Figure 6: Average distance between random nearby trajectories over time for LASA-2D. Only NCDS monotonically decreases, i.e. it is the only contractive method.

To further compare our method, we consider a robotic setting where we evaluate all the methods on a real dataset of joint-space trajectories collected on a 7-DoF robot. The last column of Table 1 shows that only NCDS scales gracefully to high-dimensional joint-space data, and outperforms the baseline methods. The supplementary material includes a video showcasing the simulation environment in which each robot executes a drawing task, depicting V-shape trajectories on the table surface.

## 4.2 OBSTACLE AVOIDANCE

Unlike baselines methods, our NCDS framework also provides dynamic collision-free motions. To assess the obstacle avoidance capabilities of our approach, we conducted a couple of experiments on the LASA dataset. The results of this experiment are reported in the third panel of Figure 8. As observed, the presence of an obstacle that completely blocks the demonstrations is represented by a red circle. Remarkably, NCDS successfully reproduced safe trajectories by effectively avoiding the obstacle and ultimately reaching the target. Moreover, the last panel in Figure 8 shows the obstacle avoidance in action as the robot's end-effector navigates around the orange cylinder. These results experimentally show how the dynamic modulation matrix approach leads to obstacle-free trajectories while still guaranteeing contraction.

## 4.3 ROBOT EXPERIMENT

To demonstrate the potential application of NCDS to robotics, we conducted several experiments on a 7-DoF Franka-Emika robotic manipulator, where an operator kinesthetically teaches the robot drawing tasks. The learned dynamics are executed using a Cartesian impedance controller. The details of the implementation and the specific network architecture are provided in Appendix A.4 and A.6.4.

|  | LASA-2D | LASA-4D | LASA-8D | 7 DoF robot |
|---|---|---|---|---|
| Euclideanizing flow | $\mathbf{0.72 \pm 0.12}$ | $3.23 \pm 0.34$ | $10.22 \pm 0.40$ | $5.11 \pm 0.30$ |
| Imitation flow | $0.80 \pm 0.24$ | $\mathbf{0.79 \pm 0.22}$ | $4.69 \pm 0.52$ | $2.63 \pm 0.37$ |
| SEDS | $1.60 \pm 0.44$ | $3.08 \pm 0.20$ | $4.85 \pm 1.64$ | $2.69 \pm 0.18$ |
| NCDS | $1.37 \pm 0.40$ | $0.98 \pm 0.15$ | $\mathbf{2.28 \pm 0.24}$ | $\mathbf{1.18 \pm 0.16}$ |

Table 1: Average dynamic time warping distances (DTWD) between different approaches. NCDS exhibits comparable performance in low dimensions, while clearly being superior in high dimensions.

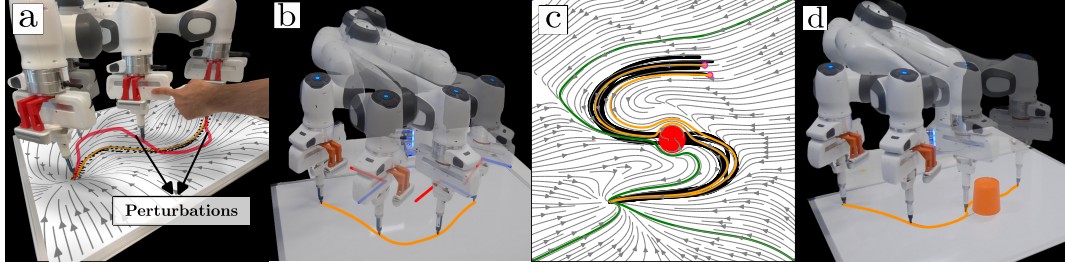

Figure 8: Robot and obstacle avoidance. Left two panels show robot experiments with orange and red paths illustrating unperturbed and perturbed motion. The first panel shows a learned vector field in $\mathbb{R}^3$ with a constant orientation. In the second panel, the vector field expands to $\mathbb{R}^3 \times \mathcal{SO}(3)$, aligning the end-effector's orientation with the motion direction. Superimposed robot images depict frames of executed motion. The right two panels display how an obstacle locally reshapes the learned vector field using the modulation matrix. In the third panel, gray contours represent the learned vector field, black trajectories depict the demonstrations, and orange/green trajectories are path integrals starting from both initial points of the demonstrations and plot corners. Magenta and red circles indicate the initial points of the path integrals and the obstacle, correspondingly. The last panel shows how the robot successfully avoids the obstacle (orange cylinder).

Firstly, we demonstrated 7 sinusoidal trajectories while keeping the end-effector orientation constant, which means that the robot end-effector dynamics only evolved in $\mathbb{R}^3$. As Fig. 8-a shows, the robot was able to successfully reproduce the demonstrated dynamics by following the learned vector field encoded by NCDS. Importantly, we also tested the extrapolation capabilities of our approach by introducing physical perturbations to the robot end-effector, under which the robot satisfactorily adapted and completed the task (see the video attached to the supplementary material).

The next experiment was aimed at testing our extension to handle orientation dynamics by learning dynamics evolving in $\mathbb{R}^3 \times \mathcal{SO}(3)$, i.e. full-pose end-effector movements. We collected several V-shape trajectories on a table surface in which the robot end-effector always faces the direction of the movement, as shown in Fig. 8-b. To evaluate the performance of this setup on a more complex dataset, we generated a synthetic LASA dataset in $\mathbb{R}^3 \times \mathcal{SO}(3)$. The results show that NCDS adeptly learns a contractive dynamical system from this dataset (see Appendix A.6.1). Another set of experiment was aimed at validating our approach on joint-space dynamic motions in a simulation environment (Corke & Haviland, 2021). These results show that among the baseline methods, NCDS outperforms others in generating motion within the joint space. (see Appendix A.6.6). Furthermore, we have further extended the joint space experiments to learning human motion skills. The findings indicate that NCDS successfully learns a contractive dynamical system from complex demonstrations in high-dimensional spaces. (see Appendix A.6.2).

## 5 DISCUSSION

In this paper, we proposed the first mechanism for designing neural networks that are contractive. This allows us to build a flexible class of models, called *neural contractive dynamical systems (NCDS)*, that can learn dynamical systems from demonstrations while retaining contractive stability guarantees. Since contraction is built into the neural network architecture, the resulting model can be fitted directly to demonstration data using standard unconstrained optimization. To cope with the complexities of high-dimensional dynamical systems, we further developed a variant of VAEs that ensures latent contrastive dynamical systems decode to contractive systems. Empirically, we found that the latent representation tends to behave simpler than the original demonstrations, which further increases the robustness of latent NCDS. We have further extended NCDS to the Lie group consisting of rotation matrices in order to model real-world robotic motions. Finally, we showed that our approach can be adapted to avoid obstacles while remaining contractive using the modulation techniques from Huber et al. (2022).

**Limitations.** Empirically, we find that NCDS is sensitive to the choice of numerical integration scheme when solving Eq. 4. Specifically, it is important to use integration schemes with adaptive step sizing for the method to be reliable. In all experiments, we used the standard DOPRI method. One concern with adaptive step sizing is that computation time is not fixed, which may negatively influence real-time implementations. On a related note, the need for numerical integration does increase the running time associated with our neural architecture compared to standard feedforward networks (see Appendix A.6.3). We, however, consider that a small price for stability guarantees.

ACKNOWLEDGMENTS

This work was supported by a research grant (42062) from VILLUM FONDEN. This project received funding from the European Research Council (ERC) under the European Union's Horizon 2020 research and innovation programme (grant agreement 757360). The work was partly funded by the Novo Nordisk Foundation through the Center for Basic Machine Learning Research in Life Science (NNF20OC0062606).

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

# A APPENDIX

## A.1 NEGATIVE DEFINITENESS OF A SYMMETRIC MATRIX

Let $\boldsymbol{A}$ be a real $n \times n$ symmetric matrix, and let $\boldsymbol{A}_S = \frac{1}{2}(\boldsymbol{A} + \boldsymbol{A}^\top)$ denote its symmetric part. Suppose that $\boldsymbol{A}_S$ is negative definite, which means that for all nonzero vectors $\boldsymbol{v} \in \mathbb{R}^n$, we have $\boldsymbol{v}^\top \boldsymbol{A}_S \boldsymbol{v} < 0$. Now let $\boldsymbol{w}$ be an eigenvector of $\boldsymbol{A}$ with corresponding eigenvalue $\lambda$, so that $\boldsymbol{A}\boldsymbol{w} = \lambda \boldsymbol{w}$. Then we have $\boldsymbol{w}^\top \boldsymbol{A}\boldsymbol{w} = \boldsymbol{w}^\top \lambda \boldsymbol{w} = \lambda \boldsymbol{w}^\top \boldsymbol{w} = \lambda |\boldsymbol{w}|^2$. Taking the transpose of the equation $\boldsymbol{A}\boldsymbol{w} = \lambda \boldsymbol{w}$, we have $\boldsymbol{w}^\top \boldsymbol{A}^\top = \lambda \boldsymbol{w}^\top$. Multiplying on the left by $\boldsymbol{w}$ and using the fact that $\boldsymbol{A}$ is symmetric, we obtain $\boldsymbol{w}^\top A \boldsymbol{w} = \lambda \boldsymbol{w}^\top \boldsymbol{w}$. Therefore, we have $\boldsymbol{w}^\top \boldsymbol{A}\boldsymbol{w} = \boldsymbol{w}^\top \boldsymbol{A}_S \boldsymbol{w} + \boldsymbol{w}^\top (\boldsymbol{A} - \boldsymbol{A}_S) \boldsymbol{w} \leq \boldsymbol{w}^\top \boldsymbol{A}_S \boldsymbol{w} < 0$. Since $\boldsymbol{w}^\top \boldsymbol{A}\boldsymbol{w} < 0$ for any nonzero eigenvector $\boldsymbol{w}$ of $\boldsymbol{A}$, we conclude that $\boldsymbol{A}$ is negative definite. Thus, the negative definiteness of the symmetric part $\boldsymbol{A}_S$ implies the negative definiteness of $\boldsymbol{A}$.

## A.2 LIE OPERATORS OF $\mathcal{SO}(3)$

To map back and forth between the Lie group $\mathcal{SO}(3)$ and its associated Lie algebra $\mathfrak{so}(3)$, we employ the logarithmic $\text{Log} : \mathcal{SO}(3) \to \mathfrak{so}(3)$, and exponential $\text{Exp} : \mathfrak{so}(3) \to \mathcal{SO}(3)$ maps, which are defined as follows,

$$\text{Exp}([\boldsymbol{r}]_\times) = \mathbb{I} + \frac{\sin(\zeta)}{\zeta}[\boldsymbol{r}]_\times + \frac{1 - \cos(\zeta)}{\zeta^2}[\boldsymbol{r}]_\times^2, \tag{9}$$

$$\text{Log}(\boldsymbol{R}) = \zeta \frac{\boldsymbol{R} - \boldsymbol{R}^\top}{2\sin(\zeta)}, \tag{10}$$

where $\zeta = \arccos\left(\frac{\text{Tr}(\boldsymbol{R})-1}{2}\right)$. Moreover, $\boldsymbol{R}$ represents the rotation matrix, and $[\boldsymbol{r}]_\times$ denotes the skew-symmetric matrix associated with the coefficient vector $\boldsymbol{r}$.

## A.3 OBSTACLE AVOIDANCE VIA MATRIX MODULATION

We here detail the used matrix modulation technique, due to Huber et al. (2022), which we use for obstacle avoidance. This approach locally reshapes the learned vector field in the proximity of obstacles, while preserving contraction guarantees. To avoid obstacles effectively, it is imperative to know both the position and geometry of the obstacle. This makes obstacle avoidance on the VAE latent space particularly difficult as we need to map the obstacle location and geometry to the latent space $\mathcal{Z}$. Hence, we directly apply the modulation matrix to the data space $\mathcal{X}$ of the decoded dynamical system.

Formally, given the modulation matrix $\boldsymbol{G}$, we can reshape the data space vector field to dynamically avoid an obstacle as follows,

$$\dot{\boldsymbol{x}} = \boldsymbol{G}(\boldsymbol{x})\boldsymbol{J}_{\mu_\xi}(\boldsymbol{z})\dot{\boldsymbol{z}}, \quad \text{with} \quad \boldsymbol{G}(\boldsymbol{x}) = \boldsymbol{E}(\boldsymbol{x})\boldsymbol{D}(\boldsymbol{x})\boldsymbol{E}(\boldsymbol{x})^{-1}, \tag{11}$$

where $\boldsymbol{E}(\boldsymbol{x})$ and $\boldsymbol{D}(\boldsymbol{x})$ are the basis and diagonal eigenvalue matrices computed as,

$$\boldsymbol{E}(\boldsymbol{x}) = [r(\boldsymbol{x})\, \mathbf{e}_1(\boldsymbol{x}) \dots \mathbf{e}_{d-1}(\boldsymbol{x})], \quad \text{and} \quad \boldsymbol{D}(\boldsymbol{x}) = \text{diag}(\lambda^r(\boldsymbol{x})\lambda^e(\boldsymbol{x}), \dots, \lambda^e(\boldsymbol{x})), \tag{12}$$

where $r(\boldsymbol{x}) = \frac{\boldsymbol{x} - \boldsymbol{x}_r}{\|\boldsymbol{x} - \boldsymbol{x}_r\|}$ is a reference direction computed w.r.t. a reference point $\boldsymbol{x}_r$ on the obstacle, and the tangent vectors $\mathbf{e}_i$ form an orthonormal basis to the gradient of the distance function $\Gamma(\boldsymbol{x})$ (see (Huber et al., 2022) for its full derivation). Moreover, the components of the matrix $\boldsymbol{D}$ are defined as $\lambda^r(\mathbf{x}) = 1 - \left(\frac{1}{\Gamma(\mathbf{x})}\right)^{\frac{1}{\rho}}$, $\lambda^e(\mathbf{x}) = 1 + \left(\frac{1}{\Gamma(\mathbf{x})}\right)^{\frac{1}{\rho}}$, where $\rho \in \mathbb{R}_+$ is a reactivity factor. Note that the matrix $\boldsymbol{D}$ modulates the dynamics along the directions of the basis defined by the set of vectors $r(\boldsymbol{x})$ and $\mathbf{e}(\boldsymbol{x})$. As stated in (Huber et al., 2022), the function $\Gamma(\cdot)$ monotonically increases w.r.t the distance from the obstacle's reference point $\mathbf{x}_r$, and it is, at least, a $C^1$ function. Importantly, the modulated dynamical system $\dot{\boldsymbol{x}} = \boldsymbol{G}(\boldsymbol{x})\boldsymbol{J}_{\mu_\xi}(\boldsymbol{z})\dot{\boldsymbol{z}}$ still guarantees contractive stability, which can be proved by following the same proof provided in (Huber et al., 2019, App. B.5).

## A.4 IMPLEMENTATION DETAILS

Both the Variational Autoencoder (VAE) and the Jacobian network $\boldsymbol{J}_\theta$ were implemented using the PyTorch framework (Paszke et al., 2019). For the VAE, we employed an injective generator based

---

**Algorithm 1:** Neural Contractive Dynamical Systems (NCDS): Training in task space

---

**Data:** Demonstrations: $\tau_n = \{(\boldsymbol{x}_t, \boldsymbol{R}_t)\}_n, n \in [1, N], t \in [1, T_n]$
**Result:** Learned contractive dynamical system
$\boldsymbol{r}_{n,t} = \text{Log}(\boldsymbol{R}_{n,t})$; // Obtaining skew-symmetric coefficients via the Lie algebra
$\boldsymbol{p}_{n,t} = [\boldsymbol{x}_{n,t}, \boldsymbol{r}_{n,t}]$; // Create new state vector
$\text{argmin}_{\boldsymbol{\xi}} \mathcal{L}_{\text{ELBO}}(\boldsymbol{\xi}^*; \boldsymbol{p}_{n,t})$; // Train the VAE
$\boldsymbol{z}_{n,t} = \mu_{\widetilde{\boldsymbol{\xi}}}^{-1}(\boldsymbol{p}_{n,t})$; // Encode all poses using the trained VAE
$\dot{\boldsymbol{z}}_{n,t} = \frac{\boldsymbol{z}_{n,t+1} - \boldsymbol{z}_{n,t}}{\Delta t}$; // Estimate latent velocities via finite differences
$\text{argmin}_{\boldsymbol{\theta}} \mathcal{L}_{\text{Jac}}(\boldsymbol{\theta}^*; \boldsymbol{p}_{n,t})$; // Train the Jacobian network

---

**Algorithm 2:** Neural Contractive Dynamical Systems (NCDS): Robot Control Scheme

---

**Data:** Current state of the robot end-effector at time $t$: $[\boldsymbol{x}_t, \boldsymbol{R}_t]$
**Result:** Velocity of the end-effector at current time step $\dot{\boldsymbol{x}}_t$
$\boldsymbol{r}_t = \text{Log}(\boldsymbol{R}_t)$; // Obtaining skew-symmetric coefficients
$\boldsymbol{p}_t = [\boldsymbol{x}_t, \boldsymbol{r}_t]$; // Create new state vector
$\boldsymbol{z}_t = \mu_{\widetilde{\boldsymbol{\xi}}}^{-1}(\boldsymbol{p}_t)$; // Compute the latent state
$\hat{\dot{\boldsymbol{z}}}_t = f(\boldsymbol{z}_t)$; // Compute the latent velocity
$\boldsymbol{J}_{\mu_{\boldsymbol{\xi}}}(\boldsymbol{z}_t) = \frac{\partial \mu_{\boldsymbol{\xi}}}{\partial \boldsymbol{z}_t}$; // Compute the Jacobian of the decoder
$\dot{\boldsymbol{x}}_t = \boldsymbol{J}_{\mu_{\boldsymbol{\xi}}}(\boldsymbol{z}_t)\hat{\dot{\boldsymbol{z}}}_t$; // Compute input space velocity

---

on M-flows (Brehmer & Cranmer, 2020), specifically using rational-quadratic neural spline flows. These flows were structured with three coupling layers. Within each coupling transform, half of the input values underwent elementwise transformation using a monotonic rational-quadratic spline. The parameters of these splines were determined through a residual network comprising two residual blocks, with each block consisting of a single hidden layer containing 30 nodes. Throughout the network, we employed $\text{Tanh}$ activations and did not incorporate batch normalization or dropout techniques. The rational-quadratic splines were constructed with ten bins, evenly distributed over the range of $(-10, 10)$. The Jacobian network was implemented using a neural network architecture consisting of two hidden layers, with each layer containing 500 nodes. The output of this network was reshaped to form a square matrix. To facilitate the integration process, we used an off-the-shelf numerical integrator called `'odeint'` from the `torchdiffeq` Python package (Chen, 2018). This package provides efficient implementations of various numerical integration methods. In our experiments, we specifically utilized the Runge-Kutta and dopri5 integration methods, which are well-known and widely used for solving ordinary differential equations.

**Loss functions:** The VAE is trained using the standard evidence lower bound (ELBO):

$$\mathcal{L}_{ELBO} = \mathbb{E}_{q_{\boldsymbol{\xi}}(\boldsymbol{z}|\boldsymbol{x})}\left[\log(p_{\phi}(\boldsymbol{x}|\boldsymbol{z}))\right] - \text{KL}\left(q_{\boldsymbol{\xi}}(\boldsymbol{z}|\boldsymbol{x})||p(\boldsymbol{z})\right), \quad (13)$$

where KL denotes the Kullback-Leibler divergence. Thereafter, the Jacobian network is trained according to the loss function $\mathcal{L}_{\text{Jac}}$, which measures the error between the observed and approximated next state,

$$\mathcal{L}_{\text{Jac}} = \|\boldsymbol{z}_{t+1} - (\boldsymbol{z}_t + \hat{\dot{\boldsymbol{z}}}_t)\|^2, \quad (14)$$

where $\boldsymbol{z}_t$, $\boldsymbol{z}_{t+1}$, and $\hat{\dot{\boldsymbol{z}}}_t$ represent the current and next observed latent states, along with the calculated latent velocity.

**Algorithms:** In this section, we elaborate on the training and robot control schemes for NCDS. The steps for training the Variational Autoencoder (VAE) and Jacobian network are outlined in Algorithm 1. Furthermore, the steps for employing NCDS to control a robot are detailed in Algorithm 2. As the algorithms show, the traning of the latent NCDS is not fully end-to-end. In the latter case, we first train the VAE (end-to-end), and then train the latent NCDS using the encoded data.

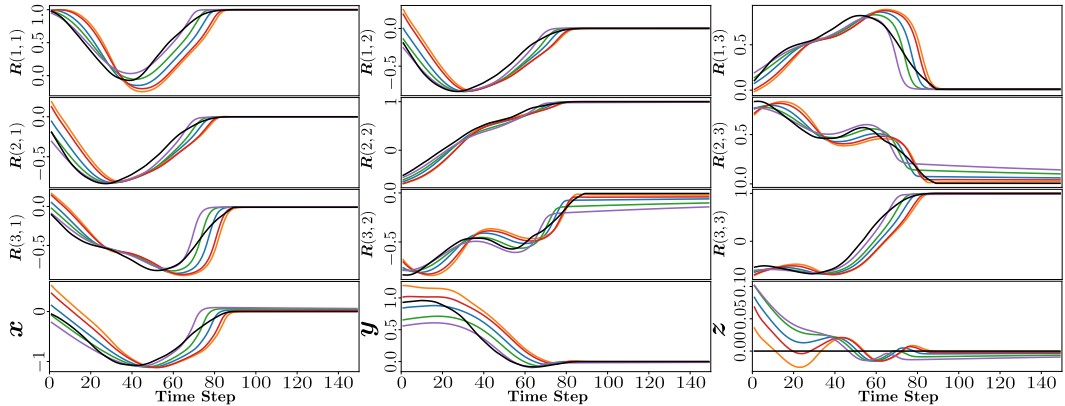

Figure 11: Time series of demonstrations and path integrals in $\mathbb{R}^3 \times \mathcal{SO}(3)$

## A.5 An Injective Decoder Architecture over the $\pi$-ball $\mathcal{B}_\pi$

To ensure that a decoder neural network has outputs over the $\pi$-ball (Sec. 3.3), we introduce a simple layer. Let $f : \mathbb{R}^d \to \mathbb{R}^D$ be an injective neural network, where injectivity is only considered over the image of $f$. If we add a TanH-layer, then the output of the resulting network is the $[-1, 1]^D$ box, i.e. $\tanh(f(\boldsymbol{z})) : \mathbb{R}^d \to [-1, 1]^D$. We can further introduce the function,

$$b(\boldsymbol{x}) = \begin{cases} \frac{\|\boldsymbol{x}\|_\infty}{\|\boldsymbol{x}\|_2} \boldsymbol{x} & \boldsymbol{x} \neq \boldsymbol{0} \\ \boldsymbol{x} & \boldsymbol{x} = \boldsymbol{0} \end{cases}. \qquad (15)$$

This function smoothly (and invertible) deforms the $[-1, 1]^D$ box into the unit ball (Fig. 9). Then, the function,

$$\hat{f}(\boldsymbol{z}) = \pi b(\tanh(f(\boldsymbol{z}))), \qquad (16)$$

is injective and has the signature $\hat{f} : \mathbb{R}^d \to \mathcal{B}_\pi(D)$.

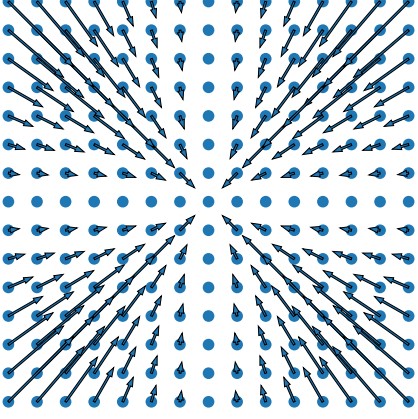

Figure 9: An illustration of the function $b(\boldsymbol{x})$ in two dimensions.

## A.6 Additional Experiments

### A.6.1 Learning $\mathbb{R}^3 \times \mathcal{SO}(3)$

The orientation trajectories of the robot experiments tend to show relatively simple dynamics. Consequently, the learning of these trajectories may not fully showcase the true potential of the approach. As a result, we conducted additional experiments to demonstrate the capacity of NCDS to encode full-pose (i.e., position-orientation) dynamic motions, which present a more challenging and comprehensive evaluation scenario. To construct a challenging dataset on $\mathcal{SO}(3)$, we projected the LASA datasets onto a 3-sphere, thereby generating synthetic quaternion data. Then we transformed this dataset to rotation matrices on $\mathcal{SO}(3)$. Later, we applied the Log map to project these points onto the Lie algebra. To construct the position data in $\mathbb{R}^3$, we simply project the 2D LASA dataset to $\mathbb{R}^3$ by concatenating a zero to the vector. Consequently, we concatenate this vector with the vector on the Lie algebra representing the orientation. This will produce a 6 dimensional state vector in $\mathbb{R}^6$. To increase the level of complexity, we employed two distinct LASA datasets for the position and orientation dimensions. Figure 10, depicts the latent dynamical system, the black dots represent the demonstration points, yellow path integrals start at the initial point of the demonstration, and green path integrals start in

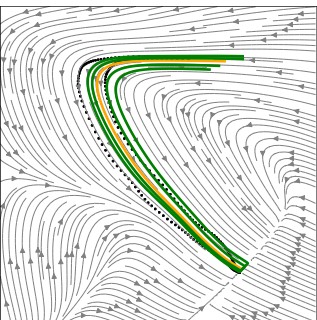

Figure 10: The background depicts the contours of the latent vector field, green and yellow curves depict the path integrals, and black dots represent demonstration in latent space.

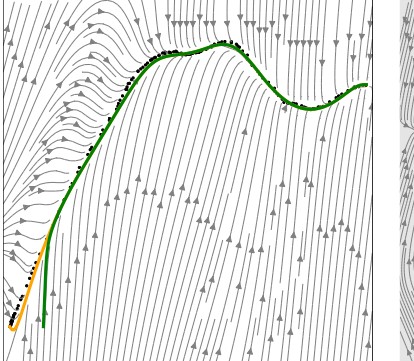 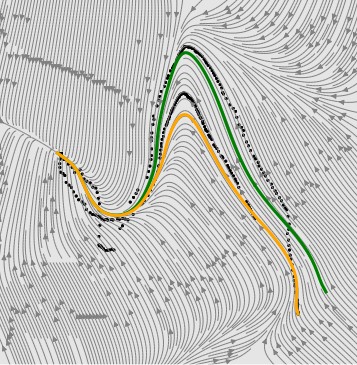

Figure 12: *Left:* Latent path integrals with NCDS trained in joint space $\mathbb{R}^{44}$. *Right:* Latent path integrals with NCDS trained on full human motion space ($\mathbb{R}^{44} \times \mathbb{R}^3 \times \mathcal{SO}(3)$). The background depicts the contours of the latent vector field, green and yellow curves depict the path integrals, and black dots represent demonstration in latent space.

random points around the demonstrations. Figure 11 shows a time series of input space trajectory in $\mathbb{R}^3$ and $\mathcal{SO}(3)$. Moreover, the black curves show the demonstration data and other trajectories show the path integrals in each dimension. The results show that NCDS is able to learn complex non-linear contractive dynamical systems in $\mathbb{R}^3 \times \mathcal{SO}(3)$.

### A.6.2 Learning human motions

To assess the model's performance in a more complex environment, we conducted a recent experiment using the KIT Whole-Body Human Motion Database (Mandery et al., 2016). This dataset captures subject-specific motions, which are standardized based on the subject's height and weight using a reference kinematics and dynamics model known as the master motor map (MMM).

**Motion in Joint space $\mathbb{R}^{44}$:** Focusing on the Tennis forehand motion, we selected this specific skill from the dataset, where each point in the motion trajectories is defined in a 44-dimensional joint space. Figure 12–*left* displays the latent contractive vector field generated by NCDS. The background depicts the contours of the latent vector field, and green and yellow curves represent latent path integrals when the initial point is respectively distant from and coincident with the initial point of the demonstration. Figure 13 illustrates how the human dummy replicates each of the generated motions. The upper row of this plot depicts the progressive evolution of the demonstrated motion from left to right. Meanwhile, the middle and bottom rows illustrate the motions generated by NCDS when the initial point is respectively distant from and coincident with the initial point of the demonstration.

**Motion in on full human motion space $\mathbb{R}^{44} \times \mathbb{R}^3 \times \mathcal{SO}(3)$:** The previous experiment has primarily concentrated on reconstructing motions within the joint space alone, which may be insufficient for some particular human motions. The data recorded from the human also includes the base (hip) link pose of the human dummy, which is essential for a complete representation of the motion. To address this, we introduced the Kick motion skill into our experiments, where each point of the motion trajectory is defined in a 44-dimensional joint space, along with the position and orientation data for the base link (hip link). This approach extends the representation of the motion to a comprehensive 50-dimensional input space.

Figure 12–*right* shows the latent path integrals generated by NCDS. Figure 14 shows the human dummy mimicking each generated motion. The upper row of this plot depicts the progressive evolution of the demonstration motion from left to right. Meanwhile, the middle and bottom rows illustrate the motions generated by NCDS when the initial point is respectively distant from and coincident with the initial point of the demonstration. This result demonstrates the inherent ability of Latent NCDS to compress a 50-dimensional nonlinear motion into a 2-dimensional space.

### A.6.3 Quantitative results

**Regularization**: It is imperative to clarify that the regularization term in Equation 2, known as epsilon $\epsilon$, plays a crucial role in defining the upper bound for the eigenvalues of the Jacobian $\hat{\boldsymbol{J}}_f(\boldsymbol{x})$,

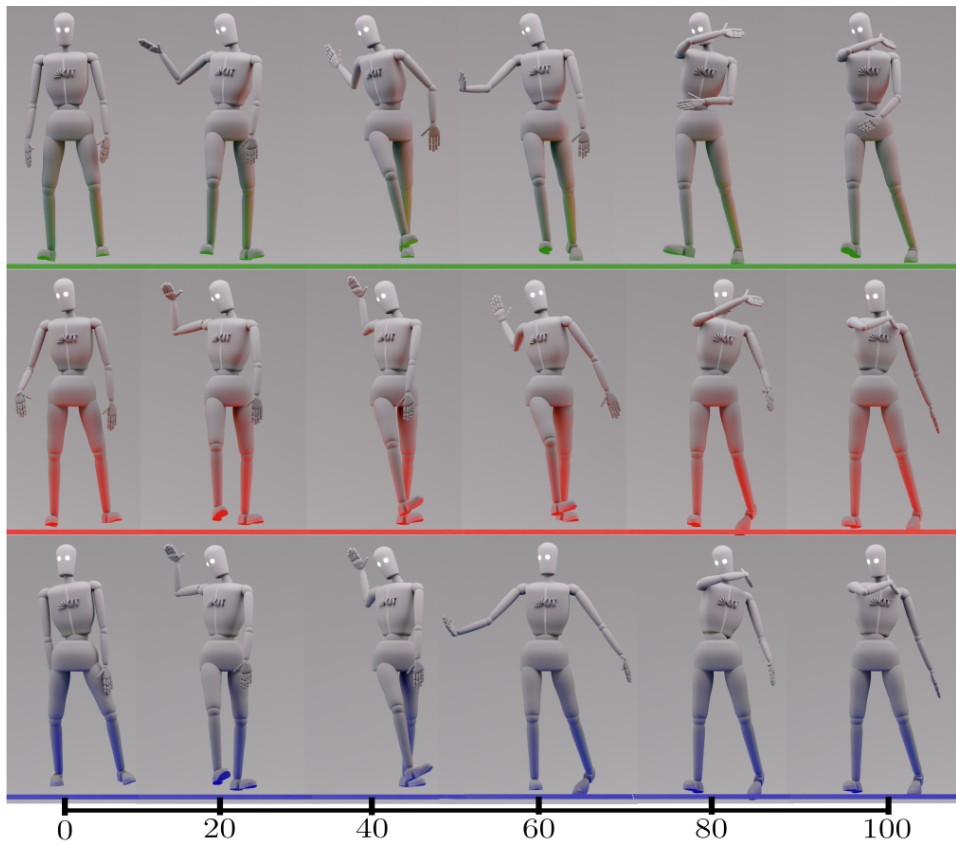

Figure 13: Generated Human Motion with NCDS trained in joint space $\mathbb{R}^{44}$: The upper row depicts the progressive evolution of the demonstration motion from left to right. Meanwhile, the middle and bottom rows illustrate the motions generated by NCDS when the initial point is respectively distant from and coincident with the initial point of the demonstration.

which inherently governs the contraction rate of the system. Figure 15 illustrates the effect of $\epsilon$ on the accuracy of the system. Adjusting this value, particularly through increments, has the potential to negatively affect the network's expressiveness, therefore compromising the reconstruction process. The observed negative effect can be attributed to the fact that this regularization is applied to the entire Jacobian matrix and not exclusively to the elements responsible for preserving the negative definiteness of the Jacobian. As a result, it might inadvertently penalize essential components of the Jacobian, leading to adverse consequences on the model's expressivity. A more targeted approach that focuses solely on the relevant elements could potentially mitigate these undesired effects and enhance the regularization's overall effectiveness.

**Unconstrained dynamical system**: To illustrate the influence of having a negative definite Jacobian in NCDS, we contrast with a system identical to NCDS except that we remove the definiteness constraint on the Jacobian network. Figure 16 shows the path integrals generated by both contractive (Figure 16-a) and unconstrained (Figure 16-b) dynamical systems. As anticipated, the dynamics generated by the unconstrained system lack stable behavior. However, the vector field aligns with the data trends in the data support regions. Furthermore, we performed similar experiments with two different baseline approaches: an MLP and a Neural Ordinary Differential Equation (NeuralODE) network, to highlight the effect of the contraction constraint on the behavior of the dynamical system. The MLP baseline uses a neural network with 2 hidden layers each with 100 neurons with $\mathrm{tanh}$ activation function. The NeuralODE baseline is implemented based on (Poli et al.), with 2 hidden layers each with 100 neurons. The model is then integrated into a NeuralODE framework. The NeuralODE is configured with the 'adjoint' sensitivity method and utilizes the 'dopri5' solver for both the forward and adjoint passes. It is important to mention that these baselines directly reproduce the velocity according to $\dot{x} = f(x)$. These models were trained with the same cost function as our NCDS's Jacobian network loss defined in Equation 14. Both networks are trained for 1000 epochs

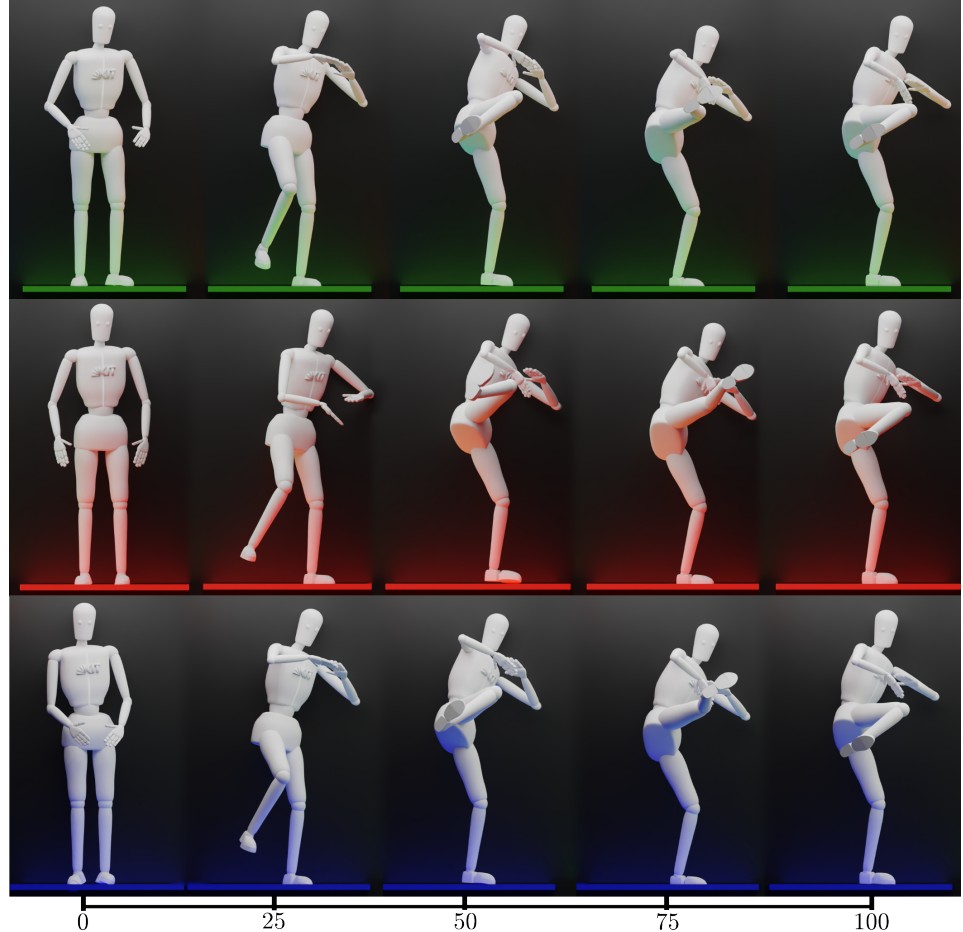

Figure 14: Generated Human Motion with NCDS trained on full human motion space ($\mathbb{R}^{44} \times \mathbb{R}^3 \times \mathcal{SO}(3)$: The upper row depicts the progressive evolution of the demonstration motion from left to right. Meanwhile, the middle and bottom rows illustrate the motions generated by NCDS when the initial point is respectively distant from and coincident with the initial point of the demonstration.

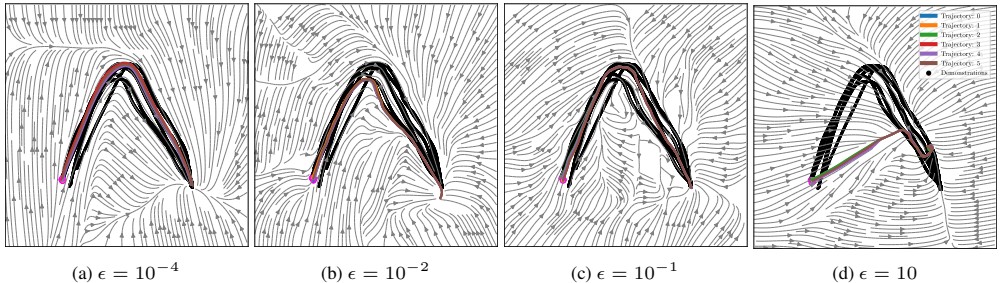

(a) $\epsilon = 10^{-4}$     (b) $\epsilon = 10^{-2}$     (c) $\epsilon = 10^{-1}$     (d) $\epsilon = 10$

Figure 15: Path integrals generated by NCDS trained using different regularization term $\epsilon$.

with ADAM optimizer. Figure 16-c and 16-d shows that both models effectively model the observed dynamics (vector field); however, as anticipated, contraction stability is not achieved.

**Integration**: Furthermore, we tested the integrator in a toy example constructed using a Sine wave under different conditions. Figure 17 shows the integrator under different conditions. Figure 17-a and -b displays cases when the absolute and relative tolerance parameters of the integrator need to be at least as $10^{-3}$. Figure 17-c shows that the number of time steps to solve the integration does not make any difference. Furthermore, Figure 17-d displays different integration methods and their data reconstruction accuracy.

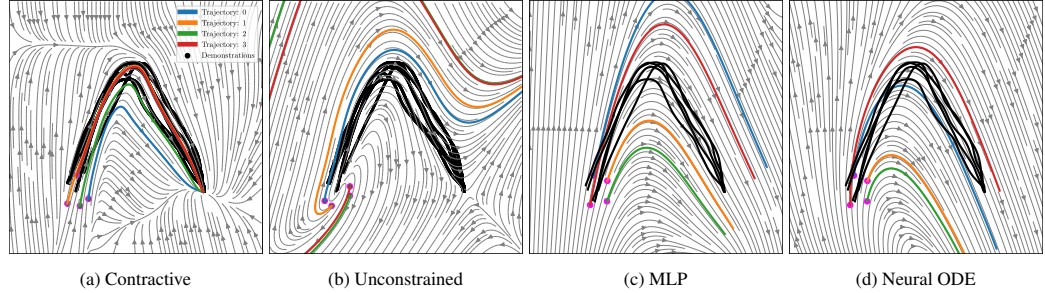

(a) Contractive      (b) Unconstrained      (c) MLP      (d) Neural ODE

Figure 16: Path integrals generated under the Neural Contractive Dynamical Systems (NCDS) setting, along with baseline comparisons using Multilayer Perceptron (MLP) and Neural Ordinary Differential Equation (NeuralODE) models.

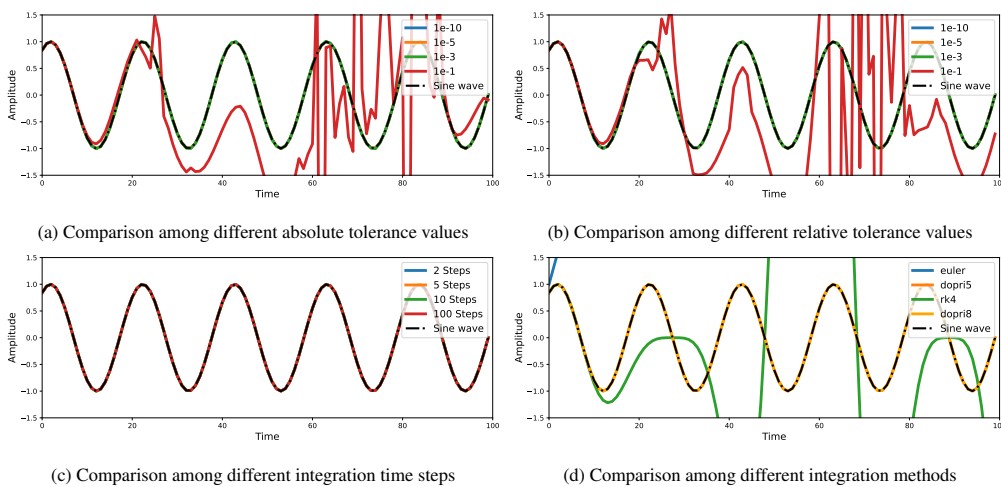

(a) Comparison among different absolute tolerance values      (b) Comparison among different relative tolerance values

(c) Comparison among different integration time steps      (d) Comparison among different integration methods

Figure 17: Evaluation of the integrator

**Activation function**: The choice of activation function certainly affects the generalization in the dynamical system (where there is no data). To show this phenomena, we ablated few common activation functions. For this experimental setup, we employed a feedforward neural network comprising two hidden layers, each consisting of 100 units. As shown in Figure 18, both the *Tanh* and *Softplus* activation functions display superior performance, coupled with satisfactory generalization capabilities. This indicates that the contour of the vector field aligns with the overall demonstration behavior outside of the data support. Conversely, opting for the *Sigmoid* activation function might yield commendable generalization beyond the confines of the data support. However, it is essential to acknowledge that this choice could potentially encounter challenges when it comes to reaching and stopping at the target.

**Dimensionality and execution time**: Table 2 compares 5 different experimental setups. First, Table 2-a reports the average execution time under the setup where the dimensionality reduction (i.e., no VAE) is discarded and the dynamical system is learned directly in the input space. As the results show, increasing the dimensionality of the input space from 2 to 8 entails a remarkable 40-fold increase in execution time for a single integration step. This empirical relationship reaffirms the inherent computational intensity tied to Jacobian-based computations. To address this issue, our approach leverages the dimensionality reduction provided by a custom Variational Autoencoder (VAE) that takes advantage of an injective generator (as decoder) and its inverse (as encoder). As showcased in Table 2-b, the VAE distinctly mitigates the computational burden. Comparing Table 2-a and Table 2-b, the considerable advantages of integrating the contractive dynamical system with the VAE into the full pipeline become evident as it leads to a significant halving of the execution time in the 8D scenario. Additionally, the findings indicate that increasing the dimensionality from 8 to 44 results in a sixfold increment in execution time. Moreover, the presented results indicate that our

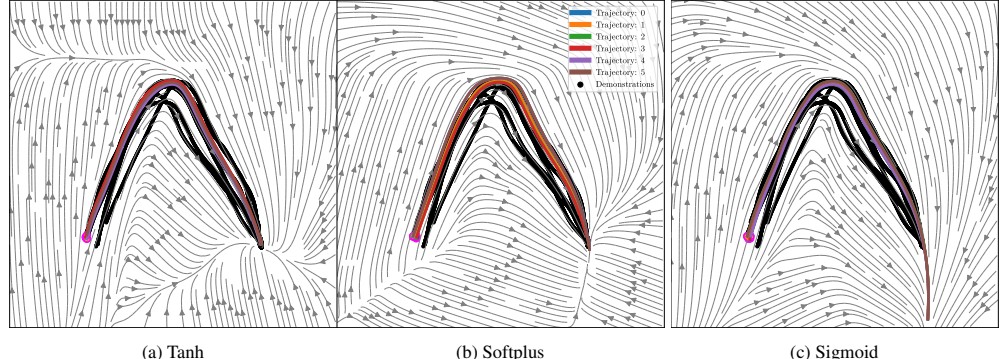

|     (a) Tanh     |   (b) Softplus   |   (c) Sigmoid   |

Figure 18: Path integrals generated by NCDS trained using different activation functions.

| Input. Dim | 2D | 8D |
|------------|------|------|
| time (ms) | 1.23 | 40.9 |

Learning the vector field directly in the input space.

| Input. Dim | 3D | 8D | 44D |
|------------|------|------|-------|
| time (ms) | 10.6 | 20.2 | 121.0 |

Learning the vector field in the latent space (full pipeline).

Table 2: Average execution time (in milliseconds) of a single integration step

current system may not achieve real-time operation. However, it is noteworthy that the video footage from our real-world robot experiments convincingly demonstrates the system's rapid query response time, sufficiently efficient to control the robot arm, devoid of any operational concerns.

### A.6.4 COMPARISONS

Table 3 summarizes the comparisons presented in the main papers in the contraction literature. Here we note a lack of consensus regarding the method or baseline for comparison in the context of learning contractive dynamical systems. A significant number of approaches have overlooked the contraction properties of their baseline, opting instead to base their comparisons solely on asymptotic stability guarantees. Following this trend, we have conducted a thorough and extensive comparative analysis involving Stable Estimator of Dynamical Systems (SEDS), Euclideanizing Flows (EF), and Imitation Flows (IF). We emphasize that neither of these baselines provides contraction guarantees but rather asymptotic stability guarantees. Therefore, the comparison is provided solely to evaluate the general stability and accuracy of the reconstructed motion.

To perform comparison between these methods and ours, we have provided two groups of datasets. The first group comprises multiple synthetic datasets, derived from the LASA dataset. The second category combines joint space trajectories of a 7-DOF Franka-Emika Panda robot arm.

**Experimental setups:**

- **Imitation Flow**: For Imitation Flow we have used a imitation flow network with the depth of 5. Each level within this depth was constructed utilizing CouplingLayer, RandomPermutation, and LULinear techniques, as recommended by the authors for managing high-dimensional data. The training process includes 1000 epochs, with a empirically chosen learning rate of $10^{-3}$. These hyper parameters were found experimentally and the best results have been selected to be compared against.

- **Euclidenizing Flow**: Here, we have used coupling network with random Fourier features parameterization, with 10 coupling layers each with 200 hidden units, and length scale of $0.45$ as the length scale for random Fourier features. The training process includes 1000 epochs, with a empirically chosen learning rate of $10^{-4}$. These hyper parameters were found experimentally and the best results have been selected to be compared against.

- **SEDS**: Here, we use 5 Gaussian functions. The training process includes 1000 epochs with MSE objective.

### A.6.5 SYNTHETIC LASA

First, we evaluate the performance and accuracy of the methods on a 2D LASA dataset as the baseline for the experiments. As depicted in Fig. 19, all methods demonstrate strong performance, effectively capturing the data trends and successfully reaching the target points.

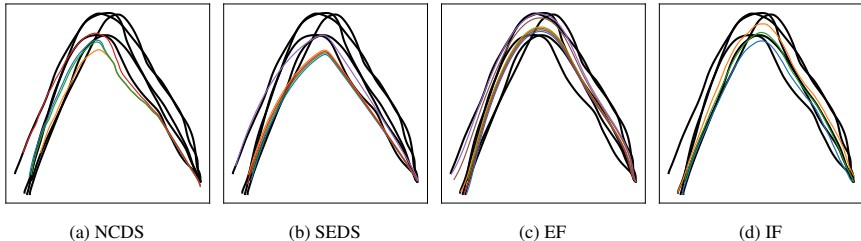

(a) NCDS      (b) SEDS      (c) EF      (d) IF

Figure 19: Path integrals generated using different approaches for the two-dimensional LASA problem. All methods provide similar reconstructions.

Examining the average time warping distance in Fig. 20, it becomes evident that EF and IF have better performance in reproducing the system dynamics.

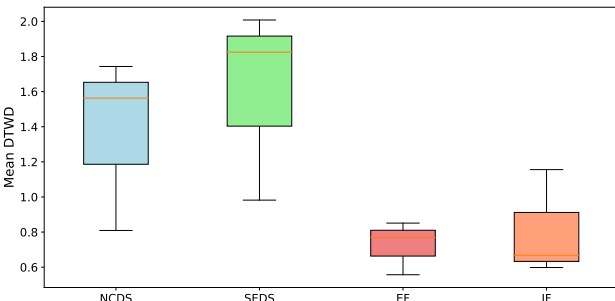

Figure 20: Average dynamic time warping distance between path integrals and the demonstrations for the two-dimensional LASA problem.

We extend our evaluation to a 4D LASA synthetic dataset created by merging two distinct datasets. In Figure 21, we observe that while NCDS and IF maintain a strong level of performance, SEDS and EF exhibit comparatively inferior results.

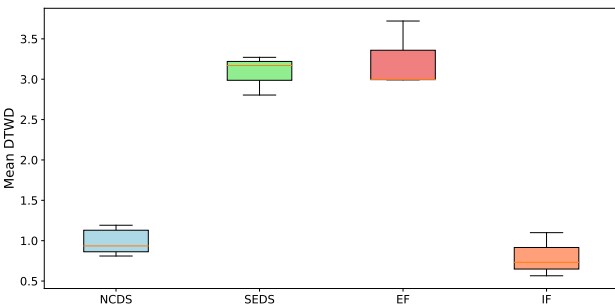

Figure 21: Average dynamic time warping distance between path integrals and the demonstrations for the four-dimensional LASA problem.

Lastly, we assess the methods using an 8D LASA synthetic dataset, constructed by concatenating 4 different datasets,

$$\mathcal{D} = [x_{\text{Angle}}, y_{\text{Angle}}, x_{\text{Line}}, y_{\text{Line}}, x_{\text{Sine}}, y_{\text{Sine}}, x_{\text{JShape}}, y_{\text{JShape}}] \in \mathbb{R}^{T \times 8}. \tag{17}$$

As illustrated in Fig. 22, NCDS has the best performance in higher-dimensional spaces.

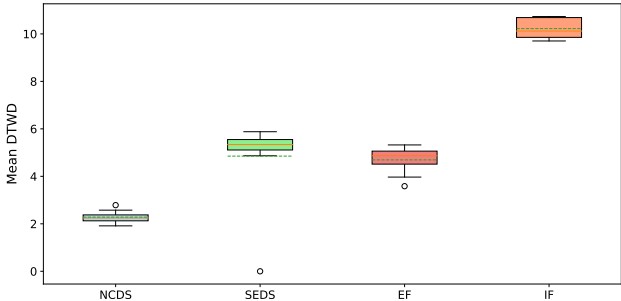

Figure 22: Average dynamic time warping distance between 5 trajectories constructed with different methods and demonstration trajectories

It is worth noting that in the 2D scenario, NCDS was directly trained in the input space, whereas in the 4D and 8D experiments, the dynamics were learned in the latent space

To make sure that we have a fair comparison we have scaled up all the methods. 1. Imitation flow: the depth of the network was scaled up by factor of 2 (from 5 to 10), We evaluated 20 different architecture with higher and lower depth and used the one with the best performance. 2. Euclideanizing Flow: the number of coupling blocks in the BijectionNet in the original code released by the Authors was increased by the factor of 2 from (10 to 20). We evaluated 20 different architecture with higher number of blocks and nodes and used the one with the best performance. 3. SEDS: we increased the number of Gaussian functions from 5 to 20. We evaluated 10 different architecture with different number of Gaussian functions used the one with the best performance. 4. NCDS: The Jacobian network did not need more parameters since the latent space has stayed 2D. Therefore, we just added the VAE to the architecture when moving from 2D to 8D.

### A.6.6 JOINT SPACE DYNAMICS

Since using the pose information including Quaternion is not feasible due to the fact that none of the methods that we compared against support non-Euclidean spaces, we have chosen to use joint space trajectories of a 7-DOF robot. Therefore, demonstration data that is used is 7-dimensional and it represents "V" shape character on a table. Fig. 23 shows the comparison on the average dynamic time warping distance (DTWD) among different methods when learning 7D joint space trajectories.

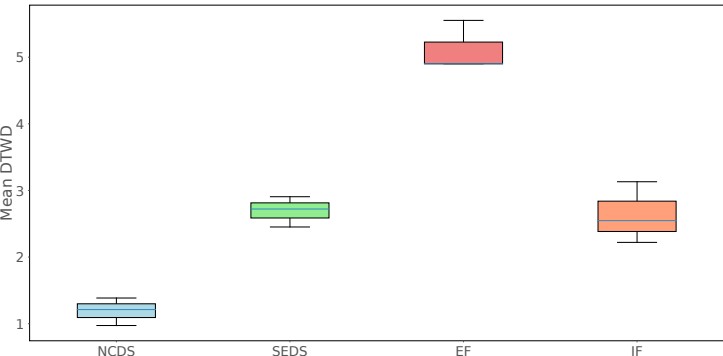

Figure 23: Average dynamic time warping distance between path integrals and the demonstrations for the 7-dimensional Joint space problem.

As shown, NCDS provides the highest reconstruction accuracy, therefore showing how our method better scales to higher-dimensional settings. The video of this experiment is int the supplementary material. Moreover, the video shows the recording process of the demonstrations and also 4 different videos from simulated robots performing the path integrals in the joint space using the four benchmarked methods.

| Paper title | Abbr. | Compared against | Year |
|---|---|---|---|
| Safe Control with Learned Certificates: A Survey of Neural Lyapunov, Barrier, and Contraction methods | | | 2022 |
| Learning Contraction Policies from Offline Data | | MPC (ILQR), RL (CQL) | 2022 |
| Neural Contraction Metrics for Robust Estimation and Control: A Convex Optimization Approach | NCM | CV-STEM, LQR | 2021 |
| Learning Stabilizable Nonlinear Dynamics with Contraction-Based Regularization | CCM-R | | 2021 |
| Learning Certified Control Using Contraction Metric | C3M | SoS (Sum-of-Squares programming), MPC, RL (PPO) | 2020 |
| Learning position and orientation dynamics from demonstrations via contraction analysis | CDSP | SEDS, CLF-DM (Control Lyapunov Function-based Dynamic Movements), Tau-SEDS, NIVF (neurally imprinted vector fields) | 2019 |
| Learning Stable Dynamical Systems using Contraction Theory | C-GMR | SEDS | 2017 |
| Learning Contracting Vector Fields For Stable Imitation Learning | CVF | DMP, SEDS and CLF-DM | 2017 |
| Learning Partially Contracting Dynamical Systems from Demonstrations | CDSP (only position) | DMP, CLF-DM | 2017 |

Table 3: The present state-of-the-art literature pertaining to the learning of contractive dynamical systems.

