# OpenReview forum: "Neural Contractive Dynamical Systems"
_ICLR.cc/2024/Conference — ICLR 2024 spotlight_

### Official Review · Reviewer_2Vh1 · 2023-10-29

**Soundness:** 3 good
**Presentation:** 4 excellent
**Contribution:** 3 good
**Rating:** 8
**Confidence:** 3

**Summary:**

### Problem Statement
The paper discusses the significant problem concerning the assurance of stability of data-driven controlling of robots, especially when the learned dynamics are controlled by neural networks. Stability is critical to prevent robots from executing harmful or undesirable actions. However, achieving global stability in dynamical systems learned from data proves challenging.

### Main Contribution
The primary contribution of this paper is the introduction of a new methodology called Neural Contractive Dynamical Systems (NCDS), which can guarantee contractive stability for the dynamics it learns, in both Euclidean and SO(3) manifolds.
This makes NCDS a highly adaptable learning architecture offering contractive stability guarantees and obstacle avoidance capabilities. The empirical results show that their approach encodes desired dynamics more accurately compared to existing state-of-the-art methods while providing stronger stability guarantees. Through NCDS, the authors aim to bridge the gap between learning robot dynamics from demonstrations and ensuring stability, which has been a notable challenge due to the extrapolating behavior of neural network models.

### Methodology
This methodology is designed to learn stable dynamical systems by constructing negative definite Jacobian from the output of neural network to ensure contraction, thus leading to global stability. The authors extend this method to address high-dimensional dynamical systems by developing a variant of the variational autoencoder using flow-based diffeomorphisms, which learns dynamics in a low-dimensional latent space while maintaining contractive stability post-decoding. They further extend their methodology to include contractive systems on the Lie group of rotations, catering to full-pose end-effector dynamic motions. The method can also be incorporated with a matrix-modulation technique to enable obstacle avoidance.

### Experiments
The paper evaluates the effectiveness and scalability of Neural Contractive Dynamical Systems (NCDS) through synthetic and real-world tasks. Initial tests on 2D trajectories from the LASA dataset demonstrated NCDS's capability in capturing and replicating underlying dynamics, even in regions not covered by original data. Compared to baseline methods like Euclideanizing flow, Imitation flow, and SEDS, NCDS was the only method showing contractive behavior indicative of stability. When scaled to higher-dimensional data (LASA-4D and LASA-8D datasets), NCDS maintained a good performance, contrary to the deteriorating performance of baseline methods.

Furthermore, the obstacle avoidance capability of NCDS was showcased on the LASA dataset, with successful generation of safe trajectories around obstacles. Real-world robot experiments on a 7-DoF Franka-Emika robotic manipulator underlined NCDS's effectiveness in reproducing demonstrated dynamics and adapting to physical perturbations. The experiments collectively underline NCDS's potential in managing various aspects of robotic motion learning, ensuring stability, and navigating obstacles, crucial for advancing real-world robotic applications.

**Strengths:**

### Originality and Significance

This is the first work to my knowledge that endows neural network based dynamics modeling with guaranteed contractive stability, which is, as the authors point out, important to robotics as people are trying to take advantage of the modeling capacity of neural networks. The conservative and non-diverging extrapolation that comes with the contractiveness enabled by this work could also benefit neural modeling of other dynamical systems apart from robotics, as the neural ordinary differential equations are known to have difficulty modeling dynamical systems when data is not enough to cover the state space, as the unregularized extrapolation of neural networks could easily lead to numerical instability during integration. Therefore this work is not only innovative but also has potentially significant impact to the community.

### Quality

The work derives the method from contraction theory and designed practical algorithms for data-driven robotics tasks, and compared with several state of the art baselines.

### Clarity

The writing of the paper is great. It excellently motivates the work and explains the method and results well with proper details.

**Weaknesses:**

1. Some implementation details are missing and the codes are not available. See more in the Questions section.

2. Limitation of baseline method choices: The baselines compared to are all focused on asymptotic stability guarantees. While these are the most relevant methods for comparison, it would be interesting to see how imitation learning methods without any stability guarantee works on the tasks, to demonstrate the necessity of stability guarantee.

**Questions:**

1. Implementation details
   1. Is training end-to-end? What are the loss functions used for training?
   2. How are the sequential data used for training? Is each trajectory used as a whole as one data point, or cut into segments as multiple data points?
   2. Second-order?

1. Multi-modality

---

> ### Author Response · Authors · 2023-11-16
>
> We are grateful to the reviewer for their interest in our work (which was excellently summarized in the review) and for valuable feedback.
>
> - *“Is training end-to-end? What are the loss functions used for training?”*
>     - **Short answer:** Training **NCDS is end-to-end**, but the **latent NCDS is not fully end-to-end**.
>     - **Detailed answer:** Training NCDS is end-to-end, but the latent NCDS is not fully end-to-end. In the latter case, we first train the VAE (end-to-end), and then train the latent NCDS using the encoded data. In more detail, the VAE is trained using the standard evidence lower bound (ELBO):
>
>         $$\\mathcal{L} _{ELBO} = \\mathbb{E} _{q _{\\mathbf{\\xi }}(\\mathbf{z}|\\mathbf{x})}\\left[\\log(p _{\\mathbf{\\phi}}(\\mathbf{x}|\\mathbf{z}))\\right] - \\mathrm{KL}\\left(q _{\\mathbf{\\xi }}(\\mathbf{z}|\\mathbf{x})||p(\\mathbf{z})\\right) , $$
>
>         where $\mathrm{KL}$ denotes the Kullback-Leibler divergence.
>
>         Thereafter, the Jacobian network is trained according to the loss function $\mathcal{L}_{\text{Jac}}$, which measures the distance between the observed and approximated next state:
>
>         $$ \\mathcal{L} _{\\text{Jac}} = \\|\\mathbf{z} _{t+1} - (\\mathbf{z} _{t} + \\hat{\\dot{\\mathbf{z}}} _t)\\|^2,$$
>
>         where $\\mathbf{z} _{t}$*, $\\mathbf{z} _{t+1}$*, and $\\hat{\\dot{\\mathbf{z}}}_t$ represent the current and next observed latent states, along with the calculated latent velocity.
>
>     - **Paper changes:** we have updated the paper and added the loss functions and the algorithms in the Appendix A.4: Implementation details.
>
> ---
>
> **Algorithm 1**: *Neural Contractive Dynamical Systems (NCDS): Training in task space*
>
> ---
>
> **Data**: Demonstrations: $\\tau _n = \\left \\{(\\mathbf{x} _t, \\mathbf{R} _t) \\right \\},  n \\in [1, N], t \\in [1, T _n]$
>
> **Result**: Learned contractive dynamical system
>
> 1. $\\mathbf{r} _{n,t} = \\operatorname{Log}(\\mathbf{R} _{n,t})$; // Obtaining skew-symmetric coefficients
> 2. $\\mathbf{p} _{n, t} = [\\mathbf{x} _{n, t}, \\mathbf{r} _{n, t}]$; *//* Create new state vector
> 3. $ \\operatorname{argmin}_\\xi \\mathcal{L} _{\\text{ELBO}}(\\xi^*; \mathbf{p} _{n,t})$; // Train the VAE
> 4. $\\mathbf{z} _{n,t} = \\mu _ {\\mathbf{\xi}} ^{\\sim 1} (\\mathbf{p} _{n,t})$; // Encode all states using the trained VAE
> 5. $\\dot{\mathbf{z}} _{n,t} = \\frac{\\mathbf{z} _{n,t+1} - \\mathbf{z} _{n,t}}{\Delta t}$; // Compute latent finite differences
> 6. $ \\operatorname{argmin}_\\theta \mathcal{L} _{\\text{Jac}}(\\theta^*; \\mathbf{p} _{n,t})$; // Train the Jacobian network
>
> ---
>
> **Algorithm 2**: Algorithm Neural Contractive Dynamical Systems (NCDS): Robot Control Scheme
>
> ---
>
> **Data:** Current state of the robot end-effector at time $t$: $[\\mathbf{x} _t, \\mathbf{R} _t]$
>
> **Result**: Velocity of the end-effector at current time step $\\dot{\\mathbf{x}} _t$
>
> 1. $\\mathbf{r} _{t} = \\operatorname{Log}(\\mathbf{R} _{t})$; // Obtaining skew-symmetric coefficients
> 2. $\\mathbf{p} _{t} = [\\mathbf{x} _{t}, \\mathbf{r} _{t}]$; // Create new state vector
> 3. $\\mathbf{z} _t = \\mu _{\\mathbf{\\xi}} ^{\\sim 1} (\\mathbf{p} _t)$; // Compute the latent state
> 4. $\\hat{\\dot{\\mathbf{z}}} _t = f(\\mathbf{z} _t)$; // Compute the latent velocity
> 5. $J_{\\mu _{\\mathbf{\\xi}}}(\\mathbf{z} _t) = \\frac{\\partial\\mu _{\\mathbf{\\xi}}}{\\partial \\mathbf{z} _t}$; // Compute the Jacobian of the decoder
> 6. $\\dot{\\mathbf{x}} _t = J _{\\mu _{\\mathbf{\\xi}}}(\\mathbf{z} _t)\\hat{\\dot{\\mathbf{z}}} _t$; // Compute input space velocity
>
> ---
>
> - “How are the sequential data used for training? Is each trajectory used as a whole as one data point, or cut into segments as multiple data points? Second-order?”
>
>    - During training, data points from all the demonstrations undergo shuffling and are then fed into the network as a batch containing $B$ data points.
>
> - *“Multi-modality”*
>     - We do not quite understand the reviewer’s question here. The term "multimodality" in the context of a dynamical system could mean different things. We are happy to do our best to answer the raised question but request a bit more detail.
> - “*Limitation of baseline method choices: The baselines compared to are all focused on asymptotic stability guarantees. While these are the most relevant methods for comparison, it would be interesting to see how imitation learning methods without any stability guarantee works on the tasks, to demonstrate the necessity of stability guarantee.”*
>     - This is a fair point. We opted for the chosen baselines as our work was really driven by a desire to provide stability. We will try to conduct these additional baseline experiments within the discussion period and report back with results later.

---

> > ### Comment · Reviewer_2Vh1 · 2023-11-20
> >
> > I thank the authors for the detailed explanation and response to the questions.
> >
> > I am also very sorry for submitting an unfinished draft for the Questions section due to a browser glitch, which caused confusion.
> >
> > In the Question 1.3, I intended to confirm whether training the Jacobian network involves second-order derivatives, as the forward propagation includes calculating Jacobian as first-order derivatives.
> >
> > In the Question 2, the full question is as follows:
> > Does contraction conflict with demonstrations that have a multimodal distribution? By "multimodal", I mean there exist multiple "modes" (potentially due to various styles or goals of the demonstrators) in the distribution of trajectories. If contractive property means prompt correction of any deviation from a template trajectory, would if ever allow for the generation of more than one modes of trajectories, as one mode of trajectories would indeed "deviate from" another mode of trajectories? For example, in the second subplot from left in the top row in Figure 5, demonstrations seem to have three branches, joining at the center point, while the generated trajectories seem to only replicate one branch when starting from the center point.
> >
> > Other than the incomplete questions mentioned above, my questions are properly addressed by the authors.

---

> > > ### Author Response · Authors · 2023-11-21
> > >
> > > - “*In the Question 1.3, I intended to confirm whether training the Jacobian network involves second-order derivatives, as the forward propagation includes calculating Jacobian as first-order derivatives.*”
> > >     - **Answer:** Good question. It is important to clarify that while the forward propagation of the Jacobian network approximates the second-order derivative of the state with respect to time (acceleration), the training process involves computing the first-order derivative with respect to the state (input) during back-propagation. This means that the training of the Jacobian network does not entail any calculation of second-order derivatives. We, however, see your point that second-order information might be made available to the optimizer through derivatives of the Jacobian network. This could potentially speed up training through faster convergence. We have, however, not explored this exciting line of thinking.
> > >
> > > ---
> > >
> > > - “*In the Question 2, the full question is as follows: Does contraction conflict with demonstrations that have a multimodal distribution? By "multimodal", I mean there exist multiple "modes" (potentially due to various styles or goals of the demonstrators) in the distribution of trajectories. If contractive property means prompt correction of any deviation from a template trajectory, would if ever allow for the generation of more than one modes of trajectories, as one mode of trajectories would indeed "deviate from" another mode of trajectories? For example, in the second subplot from left in the top row in Figure 5, demonstrations seem to have three branches, joining at the center point, while the generated trajectories seem to only replicate one branch when starting from the center point.*
> > >     - **Answer:** That is a great observation. To address this question, we can classify multimodality in two ways based on the provided clarification:
> > >         1. Multimodal trajectory distributions
> > >         2. Multimodal target distributions
> > >
> > >         In its current form, NCDS handles case 1 nicely** and this is what our reported experiment demonstrates. It is worth noting that we have made revisions to Figure 5, specifically updating the panel related to multimodal behavior. Now, upon examining the orange path integrals from the starting point of each mode, the system successfully reproduces all the modes. However, case 2 is beyond what we currently can do, and further work is needed.** Here we expect that it should be possible to do a conditional version of NCDS.
> > >
> > >     - **Revisions:** In Figure 5, the multimodal panel (first row second panel from left), has been updated to include new path integrals showing the NCDS's behavior when initiated from each mode.
> > > ---
> > >
> > > - “*Limitation of baseline method choices: The baselines compared to are all focused on asymptotic stability guarantees. While these are the most relevant methods for comparison, it would be interesting to see how imitation learning methods without any stability guarantee works on the tasks, to demonstrate the necessity of stability guarantee.*”
> > >     - **Answer:** As promised, we have performed new experiments that include imitation learning without stability guarantees. To do so, we have chosen two different baseline approaches, including MLP and Neural Ordinary Differential Equation (NeuralODE) networks. The MLP-based baseline uses a neural network with $2$ hidden layers each with $100$ neurons with $\tanh$ activation function. The Neural ODE-based baseline model is implemented based on [1], with 2 hidden layers each with 100 neurons. The model is then integrated into a NeuralODE framework. The NeuralODE is configured with the 'adjoint' sensitivity method and utilizes the 'dopri5' solver for both the forward and adjoint passes. Additionally, adjustable tolerances for the adjoint solver are set with `atol_adjoint=1e-8` and `rtol_adjoint=1e-8`. It is important to mention, that these baselines directly produce the velocity and model $\dot{x}= f(x)$. These models were trained with the same cost function as NCDS’s Jacobian network defined in Equation 14.  Both networks were trained for 1000 epochs with an Adam optimizer.  Figure 15-c and -d show that both models effectively model the observed dynamics (vector field); however, as anticipated, stability is not achieved. Similar results are reported in [2].
> > >     - **Revisions:** We have updated section **A.6.3: Unconstrained dynamical system** with the new results.
> > >
> > > [1] Poli, M., Massaroli, S., Yamashita, A., Asama, H., Park, J., & Ermon, S. "TorchDyn: Implicit Models and Neural Numerical Methods in PyTorch." https://github.com/DiffEqML/torchdyn/
> > >
> > > [2] Rodriguez, I.D., Ames, A., & Yue, Y. (2022). LyaNet: A Lyapunov Framework for Training Neural ODEs. *International Conference on Machine Learning*.

---

> > > > ### Comment · Reviewer_2Vh1 · 2023-11-23
> > > >
> > > > Thanks for the response.
> > > >
> > > > For Question 1.3, I meant that if the forward propagation involves derivatives, the back propagation through the computation graph would involve second order derivatives. This would be automatically done by the deep learning framework that is used, so it's just a minor clarifying question.
> > > >
> > > > I'm glad to see that NCDS is capable of handling multimodal trajectory distributions (with varied starting point, but not from the same starting point, which is a limitation for all deterministic approaches), and the comparison with unconstrained methods is convincing.
> > > >
> > > > I will keep my rating of 8 unchanged.

---

### Official Review · Reviewer_fKvz · 2023-10-31

**Soundness:** 3 good
**Presentation:** 2 fair
**Contribution:** 3 good
**Rating:** 6
**Confidence:** 3

**Summary:**

This work addresses the challenging problem of learning contractive dynamical systems using neural networks. The key idea is to utilize the fact that contractivity is invariant under diffeomorphisms. This motivates the use of autoencoders to learn contractive dynamics in a low-dimensional latent space. The proposed method includes a VAE architecture which naturally enforces the contractivity of the dynamics in the latent space. These results are then applied to several interesting applications, including obstacle avoidance, and learning dynamics in $SO(3)$.

**Strengths:**

The paper has several strengths.

* The idea of learning a low dimensional latent space embedding of the dynamics is interesting and novel, with a variety of interesting potential applications
* The construction of the VAE ensures that the contractivity is invariant to the mapping to the latent space (and vice versa)
* The proposed framework is extended to a variety of scenarios, including dynamics over Lie groups ($SO(3)$) and obstacle avoidance
* The paper itself is, generally speaking, well written.

**Weaknesses:**

There are a few weaknesses.

* It would be nice to have the invariance of the contractivity stated formally.
* In the discussion section (page 9), it is mentioned that the choice of integration scheme can siginificantly affect the behaviour of the learned model. This requires further discussion. For instance, how significantly does the computation time affect the performance of the model? Is there a choice of integrator that doesn't require adaptive step-sizes (perhaps a symplectic integrator)?
* It would be helpful if the training algorithm were stated explicitly in Section 3.

**Questions:**

* This work focuses on learning contractive dynamics. However, suppose the system $\dot{x} = f(x)$ is *not* contractive. What can you say about the solution to the optimization problem (i.e. the feasibility of eqns (4) and (5)) in this scenario?
* In a similar vein, it seems like this work could easily be extended to (neural) controller synthesis in the latent space. Can the authors comment on this?
* Can the authors comment on the practical effectiveness of the model in greater detail? As mentioned in the discussion section (p9), the cost of numerical integration can be extensive. Have the authors come across examples where this has been an impediment to performance?
* Could the authors also address the concerns raised in the weaknesses section?

---

> ### Author Response · Authors · 2023-11-16
>
> We appreciate the reviewer for taking the time to provide thoughtful and constructive feedback.
> - **Reviewer’s comment:** “*This work focuses on learning contractive dynamics. However, suppose the system $\dot{x} = f(x)$ is not contractive. What can you say about the solution to the optimization problem (i.e. the feasibility of eqns (4) and (5)) in this scenario?”*
>     - **Short answer:** Good question. When the system dynamics $\dot{x}=f(x)$ are not contractive, **the dynamics described by equations may integrate to arbitrary (potentially unstable) dynamics**. The system will most likely follow the trends of the training data when evaluated near the data, and then exhibit arbitrary behavior everywhere else (thereby violating stability guarantees)**.**
>     - **Detailed answer:** If the system dynamics $\dot{x}=f(x)$ are not contractive, i.e. the Jacobian of the function $f$ is not uniformly negative definite, the trajectory described by equations (4) and (5) may not exhibit stable behavior. It is worth noting that **these equations do not depend on the Jacobian being uniformly negative definite.** The optimization of equations 4 and 5 without considering the contraction formulation in equation 3 means the system might be only able to reconstruct the trajectories if started on the data support regions (not guaranteed) but the global behavior of the system outside of the data support will not be stable. **In Fig. 13 in the updated paper, you can see the differences between contractive and unconstrained dynamical systems.** As the figures show, when the resulting vector field is **not contractive, the reproduced motion follows the trend of the data only on the demonstrations region and not on the outside regions**. Overall, this system does not show any stable behavior.
>     - **Revision**: We have added a new subsection in Appendix A.6.3: Unconstrained Dynamical System discussing these results.
> ---
> - **Reviewer’s comment:** “*In a similar vein, it seems like this work could easily be extended to (neural) controller synthesis in the latent space. Can the authors comment on this?”*
>     - **Our interpretation** of this comment is the potential extension of the current work to **incorporate a controlled formulation of the dynamical system,** i.e.  $f(x,u)$, where $u$ is the control input**.** If this is correct, then **it is plausible to envision formulating the system within the latent space.** The latent space provides a compact yet semantically rich representation of the system dynamics that captures essential features of the underlying system in a more abstract and expressive manner. Extending the model to **integrate a neural controller within this latent space framework is a neat idea for control strategies at a higher level of abstraction.** The method could then explore the synergy between the latent space representation and the dynamics of the controlled system. This is a very interesting idea, but not one we have explored
> ---
> - **Reviewer’s comment:** “*Can the authors comment on the practical effectiveness of the model in greater detail? As mentioned in the discussion section (p9), the cost of numerical integration can be extensive. Have the authors come across examples where this has been an impediment to performance?*“
>     - **Short answer:** No such examples have been encountered.
>     - **Detailed answer:**  numerical integration is **one of four steps affecting performance**, where the others are: (1) encoding the current state, (2) computing the Jacobian, and (3) calculating input space velocity. Each iteration in our robot control scheme (Algorithm 2) takes up to 10 milliseconds (100﻿ Hz). Current experiments involve 7 DOF robots, encoding either pose (6-dimensional) or joint configuration  (7-dimensional), with a 2-dimensional latent space. **With respect to the cost of the numerical integration, then this depends mostly on the latent dimension and less so on the input dimension** (Table 3 in the appendix). Finally, it is worth noting that **our Python implementation is optimized for correctness rather than performance**, so there is substantial room for performance improvements.

---

> ### Author Response · Authors · 2023-11-16
>
> - **Reviewer’s comment:** “*It would be nice to have the invariance of the contractivity stated formally.”*
>     - **Short answer:** We agree that the paper was not precise enough here.
>     - **Detailed answer**: Let $\mu: \mathbb{R}^d \rightarrow \mathbb{R}^D$ be the injective decoder mapping from the low-dimensional latent space to the high-dimensional observation space. Further, let $z_t$ denote a contractive dynamical system in the latent space. Geometrically, the decoder spans a $d$-dimensional manifold embedded in $\mathbb{R}^D$ defined as $\mathcal{M} = f(\mathbb{R}^d)$. By design, the **decoder is a diffeomorphism between $\mathbb{R}^d$ and $\mathcal{M}$**; this construction is extensively discussed by Brehmer & Cranmer [1], which we build upon. **By Theorem 1 in our paper, the above implies that $f(z_t)$ is contractive.** The above further implies that the dynamical system $f(z_t)$ is along a low-dimensional manifold $\mathcal{M} \subset \mathbb{R}^D$, i.e. it does not cover all of $\mathbb{R}^D$. Such constructions are common in latent variable models. The control loop assumes that the initial robot configuration is in $\mathcal{M}$ such that it can be written $f(z_0)$**.** **Following the dynamical system moves the robot along the manifold, and the resulting motion follows a contractive system.** If the assumption is not satisfied, such that the initial configuration $x_0 \not\in \mathcal{M}$, then we have to rely on the encoder to approximate a projection onto $\mathcal{M}$, i.e. produce $z_0 = \mu^{\sim 1}(x_0)$. Note that the movement from $x_0$ to $f(z_0)$ need not be contractive, such that the distance to the nominal trajectory may increase. This is, however, a finite-time motion, after which the system is contractive.
>     - **Revisions:** In Section 3.2, we have refined and formalized the correlation between injection and contraction.
> - **Reviewer’s comment:** “*In the discussion section (page 9), it is mentioned that the choice of integration scheme can siginificantly affect the behaviour of the learned model. This requires further discussion. For instance,”*
>     - “*How significantly does the computation time affect the performance of the model?*”
>         - **Short answer:** Our findings show that as long as we keep the latent space low dimensional, the computation of the integration does not affect the overall performance of the system.
>         - **Detailed answer**: We have extensively investigated this question in Section A.6.2: *Dimensionality and execution time*, and our findings show that **as long as we keep the latent space low dimensional, the computation of the integration does not affect the overall performance** of the system. It is important to acknowledge that while maintaining a low-dimensional latent space is favorable for real-time applications, **the input space dimensionality introduces computational challenges.** Specifically, computing the Jacobian of the injective generator at each integration step becomes more resource-intensive. Furthermore, to answer this question in a more extreme scenario **we have conducted an experiment on higher-dimensional ($44D$) human motion dataset.** The new experiment showcases the performance of injective generators in encoding complex, high-dimensional data into a low-dimensional latent space ($2D$). Certainly, the considerable dimensionality of the data space inevitably impacts the runtime, with each iteration of the control loop in Algorithm 2 requiring 121 milliseconds.
>         - **Revision**: We have added a new subsection in the Appendix A.6.2 titled “Learning human motion”. We also updated Table 2 in the appendix with the new results from human motion experiments execution time.

---

> ### Author Response · Authors · 2023-11-16
>
> - **Reviewer’s comment:** “*Is there a choice of integrator that doesn't require adaptive step-sizes (perhaps a symplectic integrator)?”*
>     - **Detailed answer**: Given our limited experience with symplectic integrators, specifically designed for Hamiltonian systems, we find them less applicable to our current ODE-based system. Our research indicates that symplectic integrators excel in preserving certain properties, such as energy conservation, which are integral to Hamiltonian dynamics. Since our system does not exhibit these characteristics, opting for a symplectic integrator might introduce unnecessary complexity. On the contrary, **the Runge-Kutta 4th order (RK4) method,** **while not inherently adaptive in step size, is a fixed-step-size numerical integration technique widely employed for solving ordinary differential equations (ODEs)**. This method strikes a **balance between accuracy and computational efficiency**, particularly in ODE systems with straightforward dynamics where **stiffness is not a prominent concern.** As highlighted in Figure 13 of the paper, RK4 exhibits a certain level of accuracy, although it is not as precise as adaptive step size methods such as the Dopri5 method. This observation underscores the importance of considering the specific characteristics and requirements of our ODE system when selecting an integration method. I**n cases where our system dynamics are uncomplicated and adaptive step sizes are not imperative, RK4 remains a viable and pragmatic choice.**
> - **Reviewer’s comment:** “*It would be helpful if the training algorithm were stated explicitly in Section 3.”*
>     - **Short answer:** We agree that a training algorithm should be added to the paper.
>     - **Detailed answer:** Here, we provide detailed steps of the training and robot control schemes for NCDS. The sequential steps for training the Variational Autoencoder (VAE) and Jacobian network are listed in Algorithm 1. Simultaneously, the procedural steps for employing NCDS to control a robot are shown in Algorithm 2.
>     - **Revision**: A new subsection titled "Algorithms" has been included in the Appendix A.4, containing the respective algorithms for reference.
> ---
>
> **Algorithm 1**: *Neural Contractive Dynamical Systems (NCDS): Training in task space*
>
> ---
>
> **Data**: Demonstrations: $\\tau _n = \\left \\{(\\mathbf{x} _t, \\mathbf{R} _t) \\right \\},  n \\in [1, N], t \\in [1, T _n]$
>
> **Result**: Learned contractive dynamical system
>
> 1. $\\mathbf{r} _{n,t} = \\operatorname{Log}(\\mathbf{R} _{n,t})$; // Obtaining skew-symmetric coefficients
> 2. $\\mathbf{p} _{n, t} = [\\mathbf{x} _{n, t}, \\mathbf{r} _{n, t}]$; *//* Create new state vector
> 3. $ \\operatorname{argmin}_\\xi \\mathcal{L} _{\\text{ELBO}}(\\xi^*; \mathbf{p} _{n,t})$; // Train the VAE
> 4. $\\mathbf{z} _{n,t} = \\mu _ {\\mathbf{\xi}} ^{\\sim 1} (\\mathbf{p} _{n,t})$; // Encode all states using the trained VAE
> 5. $\\dot{\mathbf{z}} _{n,t} = \\frac{\\mathbf{z} _{n,t+1} - \\mathbf{z} _{n,t}}{\Delta t}$; // Compute latent finite differences
> 6. $ \\operatorname{argmin}_\\theta \mathcal{L} _{\\text{Jac}}(\\theta^*; \\mathbf{p} _{n,t})$; // Train the Jacobian network
>
> ---
>
> **Algorithm 2**: Algorithm Neural Contractive Dynamical Systems (NCDS): Robot Control Scheme
>
> ---
>
> **Data:** Current state of the robot end-effector at time $t$: $[\\mathbf{x} _t, \\mathbf{R} _t]$
>
> **Result**: Velocity of the end-effector at current time step $\\dot{\\mathbf{x}} _t$
>
> 1. $\\mathbf{r} _{t} = \\operatorname{Log}(\\mathbf{R} _{t})$; // Obtaining skew-symmetric coefficients
> 2. $\\mathbf{p} _{t} = [\\mathbf{x} _{t}, \\mathbf{r} _{t}]$; // Create new state vector
> 3.  $\\mathbf{z} _t = \\mu _{\\mathbf{\\xi}} ^{\\sim 1} (\\mathbf{p} _t)$;  // Compute the latent state
> 4.  $\\hat{\\dot{\\mathbf{z}}} _t = f(\\mathbf{z}  _t)$;  // Compute the latent velocity
> 5.  $J_{\\mu _{\\mathbf{\\xi}}}(\\mathbf{z} _t) = \\frac{\\partial\\mu _{\\mathbf{\\xi}}}{\\partial \\mathbf{z} _t}$;  // Compute the Jacobian of the decoder
> 6. $\\dot{\\mathbf{x}} _t = J _{\\mu _{\\mathbf{\\xi}}}(\\mathbf{z}  _t)\\hat{\\dot{\\mathbf{z}}} _t$;  // Compute input space velocity

---

> > ### Comment · Reviewer_fKvz · 2023-11-18
> > **Response to authors**
> >
> > I'd like to thank the authors for their detailed responses. You (generally) satisfactorily addressed my concerns. Based on the responses I received, as well as reading the responses to the other reviewers, I am happy to raise my score.

---

> > > ### Author Response · Authors · 2023-11-18
> > > **Thanks**
> > >
> > > Thanks for the support. In case you have further points of discussion then we are happy to engage.

---

### Official Review · Reviewer_4cHX · 2023-11-05

**Soundness:** 2 fair
**Presentation:** 3 good
**Contribution:** 3 good
**Rating:** 5
**Confidence:** 4

**Summary:**

This paper introduces a method to learn contractive dynamical systems. One method of constructing contractive systems has been to apply a diffeomorphism to provide a change of coordinates to a known contractive system. This paper proposes to use a VAE instead of a diffeomorphism, and constructs the decoder such that it is injective. As such, by enforcing that the latent dynamics is contractive, the dynamics in the data space will also be contractive. The paper also extends the method to Lie groups to account for end-effector orientation.

The paper is generally well-written, the method appears sound, and the motivations are clear. My main concerns are as follows:

1. Theorem 1 states that the contractivity is perserved under a diffeomorphism. There is a lack of analysis and formal guarentees around the assumption that an injective function acting on a lower dimensional contractive system produces a higher-D contractive system. Just as a thought experiment, what happens to coordinate points in the higher-D data space where there does not exist a coordinate in the lower-D latent space?

2. How is the collision-avoidance on the entire manipulator handled? The learned dynamical system seems to model the end-effector pose, but collision-avoidance should be handled across the body of the robot. One approach to handle this is to pull the dynamical system to the C-space of the robot and define body-points for the collision-avoidance, as done in (Zhi 2022, L4DC). This is a relevant reference and should be reviewed, as it also takes a diffeomorphic learning approach.

Overall, I believe this is a neat idea, but more clarity around the theoretical insights is needed. I'm happy to raise my score when my concerns have been address.

**Strengths:**

See above.

**Weaknesses:**

See above.

**Questions:**

See above.

---

> ### Author Response · Authors · 2023-11-16
>
> We appreciate the reviewer's engagement with our work and the valuable feedback provided.
>
> - "Theorem 1 states that the contractivity is perserved under a diffeomorphism. There is a lack of analysis and formal guarentees around the assumption that an injective function acting on a lower dimensional contractive system produces a higher-D contractive system.“
>     - **Short answer:** We agree that the paper was not precise enough here. Shortly put, **the injective decoder is a diffeomorphism between the latent space and the image of the decoder**, which is sufficient to ensure that the resulting system is contractive. We detail this argument below.
>     - **Detailed answer**: Let $\mu: \mathbb{R}^d \rightarrow \mathbb{R}^D$ be the injective decoder mapping from the low-dimensional latent space to the high-dimensional observation space. Further, let $z_t$ denote a contractive dynamical system in the latent space. Geometrically, the decoder spans a $d$-dimensional manifold embedded in $\mathbb{R}^D$ defined as $\mathcal{M} = f(\mathbb{R}^d)$. By design, the **decoder is a diffeomorphism between $\mathbb{R}^d$ and $\mathcal{M}$**; this construction is extensively discussed by Brehmer & Cranmer [1], which we build upon. **By Theorem 1 in our paper, the above implies that $f(z_t)$ is contractive.** The above further implies that the dynamical system $f(z_t)$ is along a low-dimensional manifold $\mathcal{M} \subset \mathbb{R}^D$, i.e. it does not cover all of $\mathbb{R}^D$. Such constructions are common in latent variable models. The control loop assumes that the initial robot configuration is in $\mathcal{M}$ such that it can be written $f(z_0)$**.** **Following the dynamical system moves the robot along the manifold, and the resulting motion follows a contractive system.** If the assumption is not satisfied, such that the initial configuration $x_0 \not\in \mathcal{M}$, then we have to rely on the encoder to approximate a projection onto $\mathcal{M}$, i.e. produce $z_0 = \mu^{\sim 1}(x_0)$. Note that the movement from $x_0$ to $f(z_0)$ need not be contractive, such that the distance to the nominal trajectory may increase. This is, however, a finite-time motion, after which the system is contractive.
>     - **Revision**: In Section 3.2, we have clarified and formalized the correlation between injection and contraction.
> ---
> - “Just as a thought experiment, what happens to coordinate points in the higher-D data space where there does not exist a coordinate in the lower-D latent space?”
>     - This matches the last remarks of the detailed answer above. The model provides a contractive dynamical system on top of a $d$-dimensional manifold embedded in $\mathbb{R}^D$. **When the initial robot configuration is not on this manifold, we first have to project it there.** We approximate this with our encoder $\mu^{\sim 1}$ and note that the finite-time motion from the initial configuration to the one on the manifold need not follow a contraction. **The robot video in the supplementary material includes an example of this.** **Here, the operator often pushes the robot away from the manifold spanned by the decoder after which the robot first moves back on the manifold** and thereafter follows a stable behavior.
> ---

---

> ### Author Response · Authors · 2023-11-16
>
> - “How is the collision-avoidance on the entire manipulator handled? The learned dynamical system seems to model the end-effector pose, but collision-avoidance should be handled across the body of the robot. One approach to handle this is to pull the dynamical system to the C-space of the robot and define body-points for the collision-avoidance, as done in (Zhi 2022, L4DC). This is a relevant reference and should be reviewed, as it also takes a diffeomorphic learning approach.”
>     - **Short answer:** Thank you for bringing this to our attention; we indeed overlooked this aspect in the paper.
>     - **Detailed answer:** The paper mentioned by the reviewer [2] is indeed interesting: it ensures collision avoidance in the C-space by leveraging Diffemorphic Transforms (DTs). Specifically, it employs smooth obstacle gradients obtained from continuous maps based on occupancy information to construct a vector field. Then, it uses the natural gradient of this vector field to smoothly warp around the obstacle when approaching its boundary. This is similar to obstacle avoidance approach proposed in Beik-Mohammadi et al. [3] where the C-space obstacle avoidance is accomplished by employing a pull back metric derived from the VAE decoder. This approach formulates the C-Space obstacle avoidance in the latent space of the VAE instead of the task space. To clarify, as mentioned in the paper **our current modulation matrix is formulated in the input space of the VAE, therefore, the ability to perform multiple limb obstacle avoidance can be achieved when using joint space demonstration trajectories.** However, to handle obstacle avoidance across the robot body we need to transform the joint space information into the task space and define body-points for multiple-limb collision-avoidance. Of course, **this will be computationally expensive**. Certainly, adopting the obstacle avoidance concepts introduced in [2] or [3] **eliminate the necessity for a meticulous computation of the modulation matrix, particularly for multiple limb obstacle avoidance**. Moreover, these methods **allow for the transition of the obstacle avoidance task into the latent space, mitigating any impact on real-time performance due to dimensionality of the input space.** While these approaches show potential in refining our obstacle avoidance capabilities, it is important to note that obstacle avoidance is not the central focus of our work. Nevertheless, they play a valuable role in the broader context of exploring potential enhancements in future works.
>     - **Revisions**: We have clarified C-space obstacle avoidance in the Section 3.4.
> ---
> - **References**:
>     - [1] *Johann Brehmer and Kyle Cranmer. Flows for simultaneous manifold learning and density estimation. In Neural Information Processing Systems (NeurIPS, pp. 442–453, 2020. URL https://proceedings.neurips.cc/paper_files/paper/2020/file/051928341be67dcba03f0e04104d9047-Paper.pdf.*
>     - [2] *Weiming Zhi, Tin Lai, Lionel Ott, Fabio Ramos . Diffeomorphic Transforms for Generalised Imitation Learning. Proceedings of The 4th Annual Learning for Dynamics and Control Conference, in Proceedings of Machine Learning Research. 168:508-519 Available from https://proceedings.mlr.press/v168/zhi22a.html.*
>     - [3] Beik-Mohammadi Hadi, Hauberg Soren, Arvanitidis Georgios, Neumann Gerhard, Rozo Leonel. Reactive motion generation on learned Riemannian manifolds. The International Journal of Robotics Research. 2023;42(10):729-754. doi:10.1177/02783649231193046. https://journals.sagepub.com/doi/10.1177/02783649231193046

---

### Public Comment · ~Alexander_Davydov1 · 2023-11-15
**Clarification in experimental results**

I would like to thank the authors for this really interesting work! It is closely related to a topic that some collaborators and I are working on and we look forward to citing this paper.

I had one question on the experimental results, namely the learned vector fields in Figure 5. According to the second sentence of Section 4.1, no latent structure was used. Thus, the learned vector field $\dot{x} = f(x)$ is contractive in the 2-norm in the sense of Definition 1. Accordingly, any two trajectories $x(t), y(t)$ of the vector field satisfy $\|x(t) - y(t)\|_2 \leq e^{-\varepsilon t}\|x(0) - y(0)\|_2$, i.e., any two trajectories converge to one another exponentially fast without any overshoot (see Theorem 3.9 in [1]). Pictorially, there appear to be trajectories that start close to one another and then grow further apart before converging together near the equilibrium point. This seems to violate the condition that there is no overshoot (in the 2-norm). Are the authors applying extra invertible transformations on top of the learned dynamics to yield vector fields which are contractive in more general Riemannian metrics?

[1] F. Bullo. Contraction Theory for Dynamical Systems. Kindle Direct Publishing, 1.1 edition, 2023. ISBN 979-8836646806. URL https://fbullo.github.io/ctds.

---

> ### Author Response · Authors · 2023-11-17
>
> Thank you for your interest in our work. We answer your question below.
>
> “*I had one question on the experimental results, namely the learned vector fields in Figure 5. According to the second sentence of Section 4.1, no latent structure was used. Thus, the learned vector field is contractive in the 2-norm in the sense of Definition 1. Accordingly, any two trajectories of the vector field satisfy, i.e., any two trajectories converge to one another exponentially fast without any overshoot (see Theorem 3.9 in [1]). Pictorially, there appear to be trajectories that start close to one another and then grow further apart before converging together near the equilibrium point. This seems to violate the condition that there is no overshoot (in the 2-norm). Are the authors applying extra invertible transformations on top of the learned dynamics to yield vector fields which are contractive in more general Riemannian metrics?*”
>
> **Short Answer**: No, the observed behavior is not due to the extra invertible transformations on top of the learned dynamics.
>
> **Detailed Answer**: The observed behavior in Figure 5 is a direct result of our hyperparameter tuning, specifically the regularization parameter, a critical factor influencing the system's contraction rate. As a side note, Equation 2 in the paper characterizes the contraction. In this example, we set the regularization term (in Equation 3 of the paper) to $10^{-4}$, thereby restricting the maximum negative eigenvalue of the Jacobian of the contractive system to this small value, therefore, estimating a negative definite Jacobian. Consequently, the system exhibits a less pronounced contractive behavior, strategically designed to enhance its ability to generalize across multiple demonstrations. This deliberate choice prevents the system from abruptly converging to a single narrow region, achieving adaptability and robust performance.  The rationale behind this line of thinking is extensively discussed in Appendix A.6.3 on Regularization. Moreover, we successfully experimented with increasing this parameter, resulting in a demonstrably strong contractive dynamical system, as illustrated in our new experiment detailed in Appendix A.6.2 (Figure 12). However, given the inherent nature of the task, which involves generalizing to multiple trajectories, a decision was made to opt for a smaller regularization. It is worth noting, that the regularization in Equation 3 in the paper affects all the diagonal values of the Jacobian matrix (also the values that are already negative), therefore, a more targeted approach may improve the results.
>
> Additionally, a comparable behavior can be seen in other contractive dynamical system methodologies when subjected to similar experiments. A noteworthy instance can be found in Figure 3 in [1], illustrating the quantitative performance of the CDSP algorithm on the LASA dataset. In this context, trajectories occasionally exhibit minor divergence before converging at the so-called target, again that is not a target and only a part of the contractive region. Furthermore, this behavior can be seen in the contours of the contractive vector field in CVF approach [2] in Figure 2. Similarly, you can observe the same behavior in the C-GMR behavior in Figure 4 [3]. We believe that such experimental insight in the cited papers could be possibly explained by very low contraction rates in their learned systems.
>
> [1] Ravichandar, H.c., Dani, A. Learning position and orientation dynamics from demonstrations via contraction analysis.  *Auton Robot* 43, 897–912 (2019). https://link.springer.com/article/10.1007/s10514-018-9758-x
>
> [2] Sindhwani, Vikas & Tu, Stephen & Khansari, Mohi. (2018). Learning Contracting Vector Fields For Stable Imitation Learning. https://arxiv.org/abs/1804.04878
>
> [3] C. Blocher, M. Saveriano and D. Lee, Learning stable dynamical systems using contraction theory, *2017 14th International Conference on Ubiquitous Robots and Ambient Intelligence (URAI)*, Jeju, Korea (South), 2017, pp. 124-129, https://mediatum.ub.tum.de/doc/1364708/1364708.pdf

---

### Author Response · Authors · 2023-11-17
**Revised submission**

We have carefully considered each suggestion and incorporated the recommended changes (highlighted in blue) into the revised manuscript.

Furthermore, we have included multiple videos from the new experiments in the supplementary material. These additions provide a more comprehensive understanding of our findings. We hope that these changes address all the concerns raised by the reviewers and enhance the overall quality of the paper. Either way, we hope to continue the valuable discussions.

---

### Author Response · Authors · 2023-11-22
**Open for Last-Minute Questions**

The author-reviewer discussion period is almost over, and we thank the reviewers for their insightful feedback, which helped us to improve our paper.

We remain available to address any remaining questions during these final hours.

---

### Meta-Review · Area_Chair_kU6z · 2023-12-06

**Metareview:**

*Summary*: This paper designs a new framework for learning contraction dynamical systems $\dot{x}=f(x)$ using deep neural networks. The key idea is to utilize the invariance of contractility under diffeomorphism. This motivates using a VAE to learn neural contractive dynamics in the latent space, with a carefully designed training procedure that enforces the contractility. This framework leads to several interesting applications, including end effector obstacle avoidance in manipulation and learning dynamics in $SO(3)$.

*Strength*: (1) Very clear and clean writing, easy to follow. (2) Novel and neat idea to learn contractive dynamics in the latent space using an autoencoder. (3) The proposed learning architecture guarantees contractility and works well in high-dimensional space with multimodal trajectories. (4) Interesting applications in real-world imitation learning for manipulation.

*Weakness*: (1) Need more elaborations on computational efficiency. (2) From a robotics perspective, the baselines considered in this paper are relatively narrow.

**Justification For Why Not Higher Score:**

See the weakness part.

**Justification For Why Not Lower Score:**

See the strength part.

---

### Decision · Program_Chairs · 2024-01-16

Accept (spotlight)